# Q-BENCH: A BENCHMARK FOR GENERAL-PURPOSE FOUNDATION MODELS ON LOW-LEVEL VISION

**Haoning Wu**[1*], **Zicheng Zhang**[2*], **Erli Zhang**[1*], **Chaofeng Chen**[1], **Liang Liao**[1],
**Annan Wang**[1], **Chunyi Li**[2], **Wenxiu Sun**[3], **Qiong Yan**[3], **Guangtao Zhai**[2], **Weisi Lin**[1†]
[1]S-Lab, Nanyang Technological University, [2]Shanghai Jiaotong University, [3]Sensetime
[*]Equal contribution. [†]Corresponding author. Project Page: *https://q-future.github.io/Q-Bench*.

## ABSTRACT

The rapid evolution of Multi-modality Large Language Models (MLLMs) has catalyzed a shift in computer vision from specialized models to general-purpose foundation models. Nevertheless, there is still an inadequacy in assessing the abilities of MLLMs on **low-level visual perception and understanding**. To address this gap, we present **Q-Bench**, a holistic benchmark crafted to systematically evaluate potential abilities of MLLMs on three realms: low-level visual perception, low-level visual description, and overall visual quality assessment. *a)* To evaluate the low-level *perception* ability, we construct the **LLVisionQA** dataset, consisting of 2,990 diverse-sourced images, each equipped with a human-asked question focusing on its low-level attributes. We then measure the correctness of MLLMs on answering these questions. *b)* To examine the *description* ability of MLLMs on low-level information, we propose the **LLDescribe** dataset consisting of long expert-labelled *golden* low-level text descriptions on 499 images, and a GPT-involved comparison pipeline between outputs of MLLMs and the *golden* descriptions. *c)* Besides these two tasks, we further measure their visual quality *assessment* ability to align with human opinion scores. Specifically, we design a softmax-based strategy that enables MLLMs to predict *quantifiable* quality scores, and evaluate them on various existing image quality assessment (IQA) datasets. Our evaluation across the three abilities confirms that MLLMs possess preliminary low-level visual skills. However, these skills are still unstable and relatively imprecise, indicating the need for specific enhancements on MLLMs towards these abilities. We hope that our benchmark can encourage the research community to delve deeper to discover and enhance these untapped potentials of MLLMs.

## 1 INTRODUCTION

The emergent large language models (LLMs) such as ChatGPT and Bard, as well as their excellent open-source counterparts (*e.g.*, LLaMA (Touvron et al., 2023), MPT (Team, 2023)), have served as powerful general-purpose assistants, which opens a new era for artificial intelligence (AI) from targeting specific tasks towards general intelligence. Following the advancements of LLMs, multi-modality large language models (MLLMs), as represented by LLaVA (Liu et al., 2023b), MiniGPT-4 (Zhu et al., 2023), InstructBLIP (Dai et al., 2023), and Otter (Li et al., 2023a), have brought exciting progresses on the vision field as well. They are capable of providing robust general-level abilities on visual perception/understanding and can even seamlessly dialog and interact with humans through natural language. While such abilities of MLLMs have been explored and validated on several vision-language tasks such as image captioning (Chen et al., 2015), visual question answering (Antol et al., 2015), cross-modality grounding (Peng et al., 2023), and traditional vision tasks such as image classification or segmentation (Lai et al., 2023), most attention is paid to the high-level perception and understanding of visual contents. Meanwhile, the ability of MLLMs remains not clear on **low-level visual perception and understanding**, which play significant roles in image quality assessment (IQA) (Hosu et al., 2020; Fang et al., 2020) and its associated tasks on perceiving visual distortions (*noises, blurs*) (Su et al., 2021; Wu et al., 2023d) and other low-level attributes (*color, lighting, composition, style, etc*) (Kong et al., 2016) that may relate to aesthetics and emotions of natural photos (Murray et al., 2012) and human preferences on emerging computer-

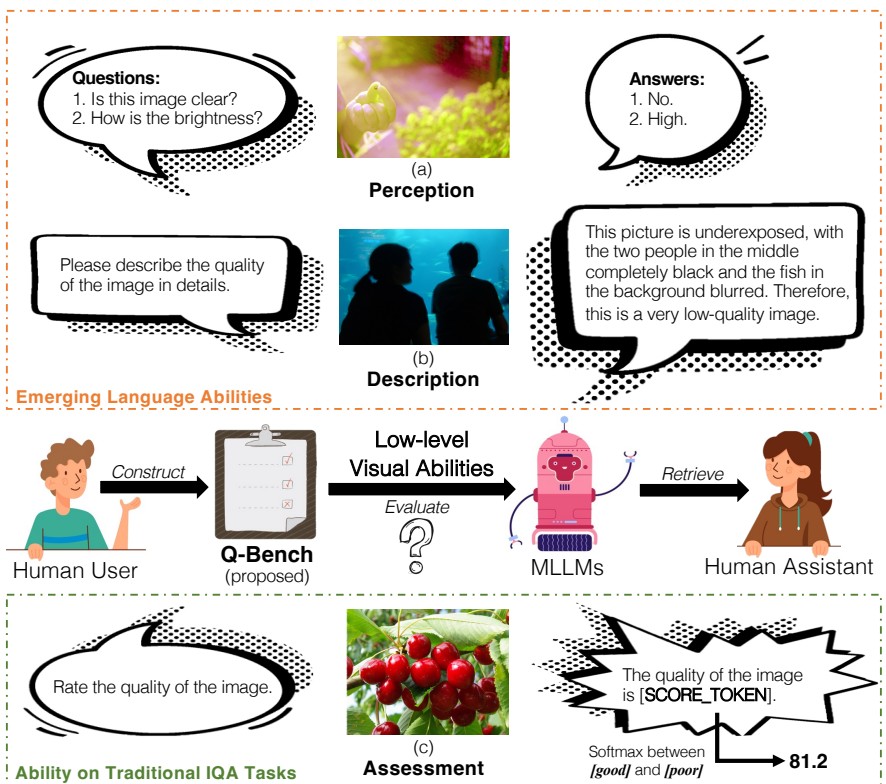

Figure 1: In the proposed **Q-Bench**, we build the first benchmark on emerging abilities of MLLMs on low-level vision, including **perception** of low-level attributes (*by correctly answering diverse queries*) and **description** of low-level quality-related information via natural language. Furthermore, the Q-bench also evaluates the quantitative **assessment** ability of MLLMs on traditional IQA tasks.

graphics generated (Zhang et al., 2023b) or AI-generated images (Li et al., 2023c; Xu et al., 2023). These low-level visual abilities are strongly associated with a wide range of applications, such as recommendation (Wu et al., 2023c), guidance on camera systems (Zhang et al., 2022), or visual quality enhancement (Zhang et al., 2018). Henceforth, it is crucial to evaluate the current abilities of these general-purpose foundation models in low-level visual perception and understanding, to ideally relieve extensive human resources to give feedback on every specific low-level task.

In our work, we propose the first systematic benchmark to measure the low-level visual perception and understanding abilities of MLLMs. Our benchmark is constructed around a key question:

*How do MLLMs emulate human ability related to low-level visual perception and understanding?*

A simple answer is **language**, which is the fundamental property of MLLMs. Specifically, we define two emerging language abilities of MLLMs on low-level vision as follows:

- *Ability 1 (A1): **Perception** of Low-level Attributes.* As shown in Fig. 1(a), like a human, an MLLM should be able to respond accurately to simple questions related to low-level attributes, *e.g* answering *'No'* for a blurry image when queried with *'Is this image clear?'*
- *Ability 2 (A2): **Description** via Natural Language.* As shown in Fig. 1(b), like a human, an MLLM should be able to describe the quality and other low-level information for an image with natural language. The descriptions should be both complete and accurate.

To systematically evaluate the low-level **perception** ability (**A1**) on various low-level attributes under diverse circumstances, we construct the **LLVisionQA** dataset, including 2,990 images from 10 diverse sources. Aligned with existing practices (Liu et al., 2023c; Lu et al., 2023), each image in LLVisionQA is equipped with a question, alongside a correct answer and false candidate answers. In LLVisionQA, we design three diverse types of questions: *Yes-or-No* questions, *What* questions, and *How* questions. Moreover, we divide low-level concerns into four quadrants, via two axes: (**1**) distortions (*blur, noises, etc*) *vs* other low-level attributes (*color, lighting, composition, etc*) (Guha et al.,

2020). **(2)** global perception (*e.g., sharpness of the whole picture*) *vs* local content-related in-context perception (*e.g., whether the red flower is in focus*) (Li et al., 2019). With three types of questions and four quadrants of concerns, the proposed **LLVisionQA** dataset provides a holistic, diverse, and balanced benchmark for the **perception** ability on low-level visual attributes of MLLMs.

For the **description** ability (**A2**), given that the output description is expected to be complex (without fixed formats), we propose the **LLDescribe** dataset by inviting experts to write long *golden* low-level descriptions (*average 58 words per description*) for 499 images, which serve as the reference texts for the single-modal GPT to evaluate MLLM output descriptions. The quality of MLLM descriptions is evaluated through three dimensions: completeness (*punish missing information*), preciseness (*punish outputs controversial with reference*), as well as relevance (*punish outputs irrelevant to low-level attributes*). With *golden* descriptions and the multi-dimensional evaluation process participated by GPT, we comprehensively evaluate the low-level description ability of MLLMs.

Besides the two emerging language abilities, we also evaluate MLLMs on the traditional IQA task, a more abstract task that requires understanding on human opinions of low-level attributes, as follows:

- *Ability 3 (A3): Precise **Assessment** Aligned with Human Opinions.* As depicted in Fig. 1(c), an MLLM should be able to predict *quantifiable* quality scores for images, which can be aligned with the human-rated mean opinion scores (MOS) on low-level visual appearances.

For the **assessment** ability (**A3**), we utilize plenty of existing IQA databases (Hosu et al., 2020; Lin et al., 2019; Li et al., 2023c) that focus on various low-level appearances of images, to benchmark MLLMs within conventional IQA settings. Specifically, we notice that MLLMs encounter difficulties in providing sufficiently quantifiable outputs, whether instructed to directly rate with texts or provide numerical outputs. To solve this challenge, we propose to extract the `softmax` pooling result on the logits of the two most frequent tokens (***good*** and ***poor***) under the response template of MLLMs (Fig 1(c)) as their quality predictions. Our studies prove that the proposed softmax-based strategy is generally better correlated with human perception than direct token outputs of MLLMs (via `argmax`), which bridges between these emergent MLLMs and the traditional IQA task settings. Under this strategy, we evaluate all MLLMs on their precise **assessment** ability by measuring the correlations between their predictions and human opinion scores in various IQA databases.

In summary, we systematically explore the potential of MLLMs on three low-level visual abilities: perception, description, and assessment. The three realms compose into the proposed **Q-Bench**, a MLLM benchmark on low-level visual tasks. Our contributions can be summarized as three-fold:

- We build a benchmark for MLLMs on low-level **perception** ability. To achieve this, we construct a first-of-its-kind balanced and comprehensive **LLVisionQA** dataset with 2,990 images with one low-level-related question-answer pair for each image. The LLVisionQA includes three question types and four quadrants of low-level concerns to ensure diversity.
- We define a benchmark process to evaluate the low-level **description** ability of MLLMs, including an **LLDescription** dataset of 499 images with expert-labelled long *golden* quality descriptions, and a GPT-assisted evaluation to rate MLLM-descriptions in terms of completeness, preciseness, and relevance compared with *golden* descriptions.
- To evaluate precise quality **assessment** ability, we propose a unified **softmax-based** quality prediction strategy for all MLLMs based on their probability outputs. With its effectiveness validated in our experiments, the proposed strategy sets up a bridge between general-purpose MLLMs and traditional IQA tasks that requires *quantifiable* scores as outputs.

## 2 CONSTRUCTING THE Q-BENCH

### 2.1 GENERAL PRINCIPLES

**Focusing on Low-level Visual Abilities of MLLMs.** Unlike existing MLLM benchmarks (Li et al., 2023b; Liu et al., 2023c; Lu et al., 2023) that aim at all-round abilities, the tasks in **Q-Bench** are constrained with two basic principles: **(1)** Requiring perception and/or understanding on low-level attributes of images; **(2)** Not requiring reasoning (*i.e. why*) or **outside** knowledge (Marino et al., 2019). We adhere to the principles in designing the **perception**, **description**, and **assessment** tasks, making the proposed **Q-bench** a focused reflection on the low-level visual abilities of MLLMs.

Table 1: Overview of the 10 diverse image source datasets in the **Q-Bench**, and the respective benchmark dataset size for each low-level ability among **perception**, **descrption** and **assessment**. The *Corrupted* COCO denotes COCO-Captions images corrupted by Michaelis et al. (2019).

| Type | Image Source Dataset | Sampled Size in **LLVisionQA** | Sampled Size in **LLDescribe** | Full Dataset Size for **Assessment** Task |
|---|---|---|---|---|
| In-the-wild | KONiQ-10K (Hosu et al., 2020) | 600 | 100 | 10,073 |
| | SPAQ (Fang et al., 2020) | 800 | 130 | 11,125 |
| | LIVE-FB (Ying et al., 2020) | 300 | 50 | 39,810 |
| | LIVE-itw (Ghadiyaram & Bovik, 2016) | 300 | 50 | 1,169 |
| Generated | CGIQA-6K (Zhang et al., 2023b) | 200 | 30 | 6,000 |
| | AGIQA-3K (Li et al., 2023c) | 198 | 30 | 2,982 |
| | ImageRewardDB (Xu et al., 2023) | 194 | 29 | *not included in* (A3) |
| Artificially-distorted | KADID-10K (Lin et al., 2019) | 81 | 20 | 10,125 |
| | LIVEMultiDistortion (Jayaraman et al., 2012) | 15 | 10 | *not included in* (A3) |
| | *Corrupted* COCO (Chen et al., 2015) | 302 | 50 | *not included in* (A3) |
| Corresponding Ability/Task in **Q-Bench** | | (A1) **Perception** | (A2) **Description** | (A3) **Assessment** |
| Total Benchmark Size for Respective Task | | 2,990 | 499 | 81,284 |

**Covering Diverse Low-level Appearances.** To cover diverse low-level appearances, we collect multi-sourced images for each task, as depicted in Tab. 1. Among all images in the **perception** and **description** tasks, *two-thirds* are in-the-wild images directly collected from social media posts, smartphones or professional photography. The rest *one-third* images are collected after various artificial distortions, or via generative processes (CGI, AIGC). Furthermore, we employ k-means clustering for the low-level attribute indicators to certify that the sub-sampled images retain high diversity. In the **assessment** task, full images of 7 IQA datasets within all three source types are evaluated through traditional IQA metrics. The diverse and multiple sources of images morph the **Q-bench** into a holistic and balanced benchmark to fairly evaluate low-level-related abilities.

## 2.2 BENCHMARK ON LOW-LEVEL **PERCEPTION** ABILITY

In the first task of Q-Bench, we evaluate the low-level **perception** ability of MLLMs to examine whether they can answer simple natural queries related to low-level attributes. For this purpose, we first collect 2,990 images (I) from multiple sources (see Table 1) with diverse low-level concerns. Then, we collect one low-level-related question (Q), one correct answer to the question (C), and 1-3 candidate false answers (F) for each image. The 2,990 (I,Q,C,F) tuples compose into the **LLVisionQA** dataset (as illustrated in Fig. 2), the first visual question answering (VQA) dataset in the low-level computer vision field. Specifically, the questions in **LLVisionQA** cover four quadrants of distinct low-level concerns (in Sec. 2.2.1) and three question types (in Sec. 2.2.2). After constructing the dataset, the (I,Q,C,F) are together fed into MLLMs for evaluation, while their outputs are further examined by GPT to judge correctness (in Sec. 2.2.3). The details are elaborated as follows.

### 2.2.1 QUADRANTS FOR LOW-LEVEL VISUAL CONCERNS

**Axis 1: Distortions *vs* Other Low-level Attributes.** The primary axis differentiates two categories of low-level perceptual attributes: **1)** technical **distortions** (Su et al., 2021), seen as the low-level characteristics that directly degrade the quality of images (Ying et al., 2020), and **2)** aesthetic-related **other low-level attributes** (Kong et al., 2016; Hou et al., 2023) which are discernible to human perception and evoke varied emotions. Several studies (Talebi & Milanfar, 2018; Ying et al., 2020; Guha et al., 2020) follow this paradigm and categorize them through a relative golden standard, that whether the attributes *directly improve or degrade picture quality* (*Yes→Distortions; No→Others*). Despite this standard, we also enumerate common types of **distortions *vs* other low-level attributes** as extra guidance for constructing the LLVisionQA dataset, as listed in Sec. A.1.2.

**Axis 2: Global Perception *vs* Local In-context Perception.** In recent research on low-level vision, it is observed that human perceptions of low-level visuals often intertwine with higher-level contextual comprehension (Li et al., 2019; Wang et al., 2021; Wu et al., 2023a). For instance, a **clear sky** might lack complex textures yet display exceptional clarity. Furthermore, localized low-level appearances can deviate from their overall counterparts, as observed by Wu et al. (2022); Ying et al. (2021). Acknowledging these differences, we curate **local in-context perception** (Fig. 2 *right*) questions, that require MLLMs to grasp the content or other context to answer correctly, while other questions are categorized as **global perception** (Fig. 2 *left*). (More analysis in Sec. A.1.2.)

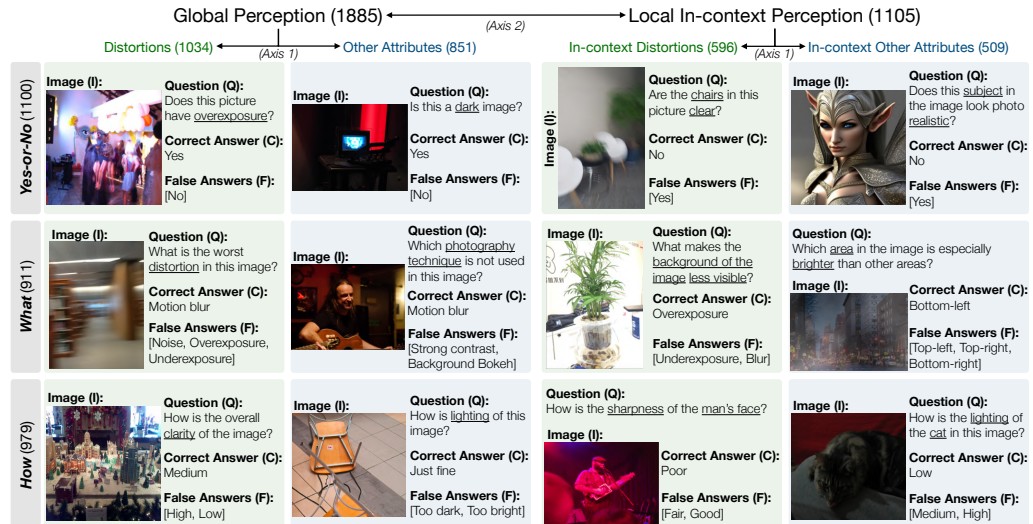

Figure 2: A dataset card of **LLVisionQA** that evaluates the low-level **perception** ability of MLLMs. 2,990 (`I,Q,C,F`) tuples are collected to cover three question types and four quadrants of low-level visual concerns, providing an all-around evaluation of low-level visual perception for MLLMs.

### 2.2.2 QUESTION TYPES

In the **LLVisionQA** dataset, we curate three question types, *Yes-or-No*, *What*, and *How* to simulate multiple query forms from humans. The details of the three question types are defined as follows.

**Type 1: *Yes-or-No* Questions.** The fundamental type of questions is *Yes-or-No*, *i.e.*, judgments. Specifically, we notice that some MLLMs especially prefer to respond with *yes* rather than *no*. To reduce such biases in our benchmark, though designing questions with answers as *yes* is easier, we ensure that around 40% of all judgments are with correct answers as *no*, via querying on **contrastive** low-level attributes or **non-existing** low-level attributes. We further measure the bias levels of different MLLMs and present a further de-biased evaluation among them, as discussed in Sec. A.3.2.

**Type 2: *What* Questions.** Despite *Yes-or-No* judgments, the *what* questions are also a common type of queries in recent MLLM benchmarks such as Lu et al. (2023). In Q-bench, they classify low-level attributes in pictures (*e.g., What distortion occurs in the image?*), or associated context given specific low-level appearances (for in-context perception questions, *e.g., Which object in the image is underexposed?*). Unlike *Yes-or-No* questions, the *What* questions examine more comprehensive low-level attribute understanding of MLLMs, by requiring correct perception on **multiple** attributes.

**Type 3: *How* Questions.** Despite the two common types, we also include a special type, the *How* questions, to cover non-extreme appearances (Wu et al., 2023d) of low-level attribute dimensions into our benchmark, as an extension to *Yes-or-No* questions. As shown in Fig. 2, we can query *How is the clarity of the image?* for the image with both clear and blurry areas, and answer with Medium. With this special question type, we broaden the Q-bench into **finer-grained** low-level perception.

### 2.2.3 GPT-ASSISTED EVALUATION PROCESS

After constructing the LLVisionQA dataset, we feed it to multiple MLLMs to evaluate their abilities on low-level visual **perception**. The input format to query MLLMs is exemplified as follows:

*#User: How is the clarity of the image?* (`Question`) *[IMAGE_TOKEN]* (`Image`)
*Choose between one of the following options: A. High* (`Correct`) *B. Medium* (`Wrong`) *C. Low* (`Wrong`)

The correct and wrong answers are shuffled during the actual evaluation. Moreover, while traditional visual question answering (Antol et al., 2015; Marino et al., 2019) tasks typically employ traditional language metrics (BLEU-4, CIDEr) to compare performance, as observed by recent studies (Ye et al., 2023) and validated by us, most MLLMs cannot consistently provide outputs on **instructed formats**. Given the question above, different MLLMs may reply *"A."*, *"High"*, *"The clarity of the image is high."*, *"The image is of high clarity."* (all correct), which are difficult to be exhaustively-

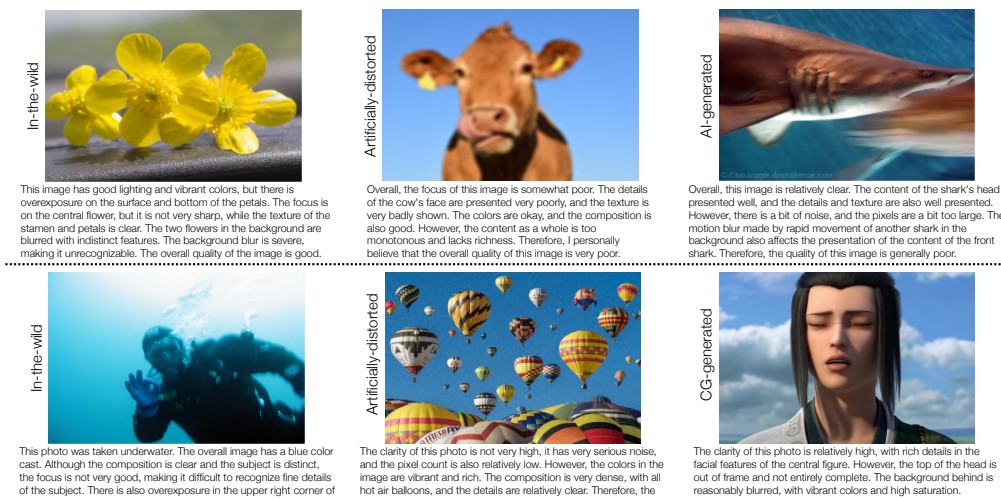

Figure 3: A dataset card of **LLDescribe** that evaluates the low-level **description** ability of MLLMs. 499 images from 10 diverse sources are labeled with *golden* descriptions, to serve as __text__ references for single-modal GPT to evaluate the completeness, preciseness, and relevance of MLLM outputs.

included under traditional metrics. To solve this problem, we design, validate, and employ a **5-round** GPT-assisted evaluation process inspired by Liu et al. (2023c). Under this process, the question, correct answers, and MLLM replies are fed into GPT for evaluation (See Sec. A.2.1 for its details).

## 2.3 BENCHMARK ON LOW-LEVEL **DESCRIPTION** ABILITY

In the second task of Q-Bench, we evaluate the language **description** ability of MLLMs on low-level information. This task is a sibling task of image captioning (Chen et al., 2015; Young et al., 2014; Agrawal et al., 2019) that describes image content with natural language, with a specific concern on the low-level appearance of images. To evaluate this ability automatically, we first derive a *golden* low-level description dataset, denoted as **LLDescribe** (Sec. 2.3.1), including one long (*average 40 words*) *golden* description provided by experts for each of 499 images. With these *golden* text descriptions, we are able to measure the quality of output low-level descriptions from MLLMs with a single-modal GPT, under the three dimensions: **completeness**, **preciseness**, as well as **relevance** (Sec 2.3.2). The discussions of the *golden* descriptions and the evaluation process are as follows.

### 2.3.1 DEFINING *Golden* LOW-LEVEL DESCRIPTIONS FOR IMAGES

For the description ability, MLLMs should accurately and completely describe low-level visual information of images. Thus, the *ground truths* for these MLLMs are also built within a basic principle to cover as many low-level concerns as possible, so long as they are enumerated in Sec. 2.2.1 and occur in images. The resulting *golden* descriptions in **LLDescribe** have an average duration of **58** words, notably longer than common high-level image caption datasets (**11** for Agrawal et al. (2019), **10** for Chen et al. (2015)). Similar to the **LLVisionQA** dataset for the perception task, the 499 images in **LLDescribe** dataset also include all 10 sources (as in Tab. 1) to cover images with diverse low-level appearances. The *golden* descriptions on different sources of images are depicted in Fig. 3.

### 2.3.2 EVALUATION WITH SINGLE-MODAL GPT

Recent studies (Zheng et al., 2023) have proved single-modal GPT (OpenAI, 2023) to be a reliable evaluation tool for pure language tasks. Via the **LLDescribe** dataset, we convert the multi-modality problem into a text-only setting, by matching the MLLM outputs with the *golden* descriptions with single-modal GPT under three dimensions: **(1) Completeness.** More matched information with the *golden* description is encouraged. **(2) Preciseness.** The controversial information with the *golden* description is punished. **(3) Relevance.** More proportions of MLLM outputs should be related to low-level information, instead of others. Each dimension is scored among [0,1,2]. Similar as Sec. 2.2.3, we repeat **5 rounds** for each single evaluation and collect the weighted average as the final score. The detailed settings for GPT to evaluate the three dimensions are in Sec. A.2.2.

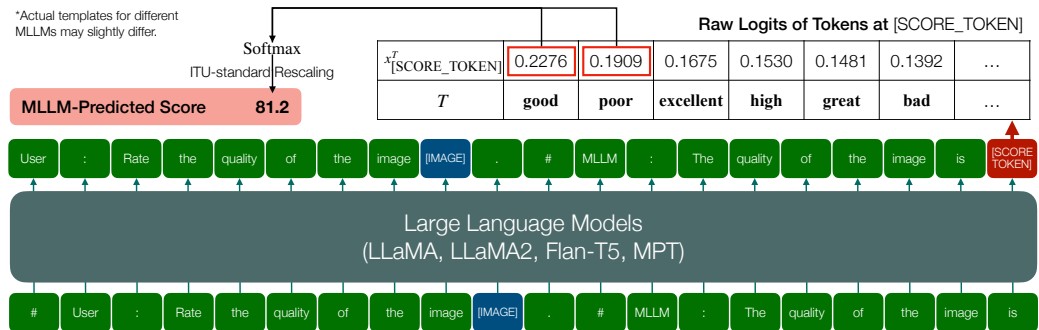

Figure 4: The proposed softmax-based quality **assessment** strategy for MLLMs. Instead of directly decoding tokens from the *[SCORE_TOKEN] position*, the strategy extracts log probabilities (logits) of ***good*** and ***poor***, and predicts *quantifiable* score via a `softmax` pooling between the two logits.

## 2.4 BENCHMARK ON PRECISE QUALITY **ASSESSMENT** ABILITY

In the third task, we benchmark the ability of MLLMs to provide *quantitative* **assessment** on the overall low-level appearance of images. Unlike the two tasks above, we utilize existing IQA datasets that are collected across a variety of low-level appearances to evaluate how MLLMs can predict quantitative quality scores **aligned with human opinions**. All the three types of IQA datasets (*in-the-wild*, *generated*, *artificially-distorted*) as mentioned in Sec. 2.1 are evaluated, to provide a broad range measurement of the assessment ability of MLLMs. Nevertheless, how to collect *quantifiable* quality scores from MLLMs remains challenging as their outputs only have weak measurability (Sec. 2.4.1). Noticing that MLLMs can provide probabilities of tokens, we employ softmax pooling on the logits of ***good*** and ***poor*** under a simple and direct prompt template, deriving into *quantifiable* quality score predicted by MLLMs (Sec. 2.4.2), as illustrated in Fig. 4. Details are as follows.

### 2.4.1 WEAK MEASURABILITY OF MLLM OUTPUTS

In Q-Bench, we aim to fairly compare the **assessment** ability between different MLLMs on diverse low-level appearances. Henceforth, our principle is to define a unified, **simplest** instruction that is applicable for all MLLMs on all IQA datasets. Under this principle, we conduct toy experiments on LLVisionQA on Shikra and LLaVA-v1, with two simple instruction strategies: **(A) Direct Instruction,** in which the prompt is designed as simple as *"Rate the quality of the image"*. The top-frequency answers are ***good*** (78%), and ***poor*** (20%), with other outputs almost negligible. **(B) Numerical Instruction,** in which we specifically instruct numerical ratings, with the prompt: *"Score the quality of the image from 1 to 5, with 1 as lowest and 5 as highest."*. Under the numerical strategy, the top-frequency answers are **5** (84%), **1** (9%), and **3** (5%); though within the score range, the frequencies of scores **2** and **4** are both less than 1%. The toy experiments imply the weak measurability of MLLM outputs, given that the answers are statistically **1)** biased towards *positive*, **2)** biased towards *extreme*, and **3)** with *only two* effective scales. Therefore, it is necessary to explore extended strategies for MLLMs to provide truly *quantifiable* outputs for low-level **assessment**.

### 2.4.2 A SOFTMAX-BASED EVALUATION STRATEGY

Given the above observations, we design the softmax-based evaluation strategy (Fig. 4) to reduce the negative impacts of the biases and lack of scales. To start with, we design our strategy within the **Direct Instruction**, which is more general and less biased than the **Numerical Instruction**. The strategy is based on the observation that two top-frequency outputs, ***good*** and ***poor***, can be considered as anchors for better and worse human perception, and the **Direct Strategy** can be approximated into a binary classification problem on the *[SCORE_TOKEN]* position, or technically, an `argmax` between the logits of ***good*** ($x^{\mathbf{good}}_{SCORE\_TOKEN}$) and ***poor*** ($x^{\mathbf{poor}}_{SCORE\_TOKEN}$) on this position. In our revised strategy, we modify the `argmax` into `softmax` to collect better *quantifiable* scores:

$$q_{\text{pred}} = \frac{e^{x^{\mathbf{good}}_{SCORE\_TOKEN}}}{e^{x^{\mathbf{good}}_{SCORE\_TOKEN}} + e^{x^{\mathbf{poor}}_{SCORE\_TOKEN}}} \tag{1}$$

This simple and generally-applicable strategy enables us to collect *quantifiable* outputs ($q_{\text{pred}}$) from MLLMs with higher correlation to human ratings, as verified in our experimental analysis (Tab. 9).

Table 2: Results on the `test` subset for the low-level **Perception** ability of MLLMs. MLLMs with *top-3* performance in each sub-category and the overall **LLVisionQA** is emphasized with **boldface**.

| Sub-categories | Question Types | | | Quadrants of Low-level Concerns | | | | |
|---|---|---|---|---|---|---|---|---|
| **Model** (*variant*) | *Yes-or-No*↑ | *What*↑ | *How*↑ | *Distortion*↑ | *Other*↑ | *In-context Distortion*↑ | *In-context Other*↑ | *Overall*↑ |
| *random guess* | 50.00% | 28.48% | 33.30% | 37.24% | 38.50% | 39.13% | 37.10% | 37.94% |
| LLaVA-v1.5 (*Vicuna-v1.5-7B*) | 64.60% | 59.22% | 55.76% | 47.98% | 67.30% | 58.90% | 73.76% | 60.07% |
| LLaVA-v1.5 (*Vicuna-v1.5-13B*) | 64.96% | **64.86%** | 54.12% | 53.55% | 66.59% | 58.90% | 71.48% | 61.40% |
| InternLM-XComposer-VL (*InternLM*) | 68.43% | 62.04% | **61.93%** | 56.81% | **70.41%** | 57.53% | **77.19%** | **64.35%** |
| IDEFICS-Instruct (*LLaMA-7B*) | 60.04% | 46.42% | 46.71% | 40.38% | 59.90% | 47.26% | 64.77% | 51.51% |
| Qwen-VL (*QwenLM*) | 65.33% | 60.74% | 58.44% | 54.13% | 66.35% | 58.22% | 73.00% | 61.67% |
| Shikra(*Vicuna-7B*) | 69.09% | 47.93% | 46.71% | 47.31% | 60.86% | 53.08% | 64.77% | 55.32% |
| Otter-v1 (*MPT-7B*) | 57.66% | 39.70% | 42.59% | 42.12% | 48.93% | 47.60% | 54.17% | 47.22% |
| InstructBLIP (*Flan-T5-XL*) | 69.53% | 59.00% | 56.17% | 57.31% | 65.63% | 56.51% | 71.21% | 61.94% |
| InstructBLIP (*Vicuna-7B*) | 70.99% | 51.41% | 43.00% | 45.00% | 63.01% | 57.19% | 64.39% | 55.85% |
| VisualGLM-6B (*GLM-6B*) | 61.31% | 53.58% | 44.03% | 48.56% | 54.89% | 55.48% | 57.79% | 53.31% |
| mPLUG-Owl (*LLaMA-7B*) | **72.45%** | 54.88% | 47.53% | 49.62% | 63.01% | **62.67%** | 66.67% | 58.93% |
| LLaMA-Adapter-V2 | 66.61% | 54.66% | 51.65% | 56.15% | 61.81% | 59.25% | 54.55% | 58.06% |
| LLaVA-v1 (*Vicuna-13B*) | 57.12% | 54.88% | 51.85% | 45.58% | 58.00% | 57.19% | 64.77% | 54.72% |
| MiniGPT-4 (*Vicuna-13B*) | 60.77% | 50.33% | 43.00% | 45.58% | 52.51% | 53.42% | 60.98% | 51.77% |
| **GPT-4V** (*Close-Source Model*) | 77.92% | 79.18% | 62.68% | 70.58% | 73.03% | 74.66% | 77.95% | 73.36% |
| *Junior-level Human* | 82.48% | 79.39% | 60.29% | 75.62% | 72.08% | 76.37% | 73.00% | 74.31% |
| *Senior-level Human* | 84.31% | 88.94% | 72.02% | 79.65% | 79.47% | 83.90% | 87.07% | 81.74% |

## 3 RESULTS ON Q-BENCH

In Q-Bench, we evaluate **15** variants on **13** up-to-date popular and competitive open-source MLLMs, together with GPT-4V, under zero-shot settings. More results and analyses are appended in Sec. A.3.

### 3.1 RESULTS AND OBSERVATIONS ON PERCEPTION

**Open-Source MLLMs.** For a holistic examination on the **perception** ability of MLLMs, we evaluate the multi-choice correctness of MLLMs on different sub-categories of the **LLVision** dataset, which is equally divided as `dev` (Tab. 7, *will be released*) and `test` (Tab. 2, *will keep private*) subsets. We are glad that the majority of MLLMs can significantly outperform *random guess* on all sub-categories. Considering that all participating MLLMs are without any explicit training on low-level visual attributes, these results show strong potentials for these general-purpose models when further fine-tuned with respective low-level datasets. Among all MLLMs, the recently-released InternLM-XComposer-VL reaches the best accuracy on this question-answering task, followed by LLaVA-v1.5, QWen-VL and InstructBLIP (*Flan-T5*), which show rather close results. By achieving **more than 60%** accuracy on both subsets, these models show exciting potentials as robust low-level visual assistants in the future. Another key observation is that almost all methods **perceive worse on distortions** than other low-level attributes. One exception is LLaMA-Adapter-V2, which is the only MLLM that adopts **multi-scale** features as visual inputs. We also notice that all MLLMs prefer *yes* than *no* among **Yes-or-No** questions, as analyzed in Tab. 8; qualitative comparisons are illustrated in Fig. 10. For Kosmos-2, we specially adopt *close-set* inference for it, as discussed in Sec. A.2.1.

**GPT-4V *vs* Human.** To evaluate the low-level **perception** abilities of the commercial MLLM, GPT-4V, we gauge its accuracy against human using the `test` subset of **LLVision** dataset. GPT-4V exhibits competitive performance and outperforms open-source MLLMs by a large margin (**+9%**), and on par accuracy with the *Junior-level Human*. Despite its prowess, there is still a way to go for GPT-4V before it can match the overall proficiency of the *Senior-level Human (with experiences on low-level visual tasks*, **8%** better than GPT-4V). Furthermore, across all categories, the results show that GPT-4V, much like its open-source counterparts, faces challenges in recognizing **distortions**.

### 3.2 RESULTS AND OBSERVATIONS ON DESCRIPTION

For the **description** ability, InternLM-XComposer-VL reaches best proficiency again, especially in terms of the relevance dimension. Nevertheless, in the perspective of the completeness and precision of the descriptions, even the best of all MLLMs cannot obtain an excellent score; on the contrary, almost all MLLMs reach an acceptable standard (0.8/2.0). In general, all MLLMs at present are only with relatively limited and primary ability to provide low-level visual descriptions. We also conduct a qualitative comparison for MLLM descriptions in Sec. A.3.3.

Table 3: Results on the low-level **Description** ability of MLLMs. $P_i$ denotes frequency for score $i$.

| Dimensions | Completeness | | | | Precision | | | | Relevance | | | | Sum.↑ |
|---|---|---|---|---|---|---|---|---|---|---|---|---|---|
| Model (variant) | $P_0$ | $P_1$ | $P_2$ | score↑ | $P_0$ | $P_1$ | $P_2$ | score↑ | $P_0$ | $P_1$ | $P_2$ | score↑ | |
| LLaVA-v1.5 (Vicuna-v1.5-7B) | 27.48% | 54.74% | 17.78% | 0.90 | 30.51% | 26.04% | 43.45% | 1.13 | 10.85% | 60.34% | 28.81% | 1.18 | 3.21 |
| LLaVA-v1.5 (Vicuna-v1.5-13B) | 27.68% | 53.78% | 18.55% | 0.91 | 25.45% | 21.47% | 53.08% | **1.28** | 6.31% | 58.75% | 34.94% | 1.29 | 3.47 |
| InternLM-XComposer-VL (InternLM) | 19.94% | 51.82% | 28.24% | 1.08 | 22.59% | 28.99% | 48.42% | 1.26 | 1.05% | 10.62% | 88.32% | **1.87** | **4.21** |
| IDEFICS-Instruct (LLaMA-7B) | 28.91% | 59.16% | 11.93% | 0.83 | 34.68% | 27.86% | 37.46% | 1.03 | 3.90% | 59.66% | 36.44% | 1.33 | 3.18 |
| Qwen-VL (QwenLM) | 26.34% | 49.13% | 24.53% | 0.98 | 50.62% | 23.44% | 25.94% | 0.75 | 0.73% | 35.56% | 63.72% | 1.63 | 3.36 |
| Shikra (Vicuna-7B) | 21.14% | 68.33% | 10.52% | 0.89 | 30.33% | 28.30% | 41.37% | 1.11 | 1.14% | 64.36% | 34.50% | 1.33 | 3.34 |
| Otter-v1 (MPT-7B) | 22.38% | 59.36% | 18.25% | 0.96 | 40.68% | 35.99% | 23.33% | 0.83 | 1.95% | 13.20% | 84.85% | 1.83 | 3.61 |
| Kosmos-2 | 8.76% | 70.91% | 20.33% | **1.12** | 29.45% | 34.75% | 35.81% | 1.06 | 0.16% | 14.77% | 85.06% | 1.85 | 4.03 |
| InstructBLIP (Flan-T5-XL) | 23.16% | 66.44% | 10.40% | 0.87 | 34.85% | 26.03% | 39.12% | 1.04 | 14.71% | 59.87% | 25.42% | 1.11 | 3.02 |
| InstructBLIP (Vicuna-7B) | 29.73% | 61.47% | 8.80% | 0.79 | 27.84% | 23.52% | 48.65% | 1.21 | 27.40% | 61.29% | 11.31% | 0.84 | 2.84 |
| VisualGLM-6B (GLM-6B) | 30.75% | 56.64% | 12.61% | 0.82 | 38.64% | 26.18% | 35.18% | 0.97 | 6.14% | 67.15% | 26.71% | 1.21 | 2.99 |
| mPLUG-Owl (LLaMA-7B) | 28.28% | 37.69% | 34.03% | 1.06 | 26.75% | 18.18% | 55.07% | **1.28** | 3.03% | 33.82% | 63.15% | 1.60 | 3.94 |
| LLaMA-Adapter-V2 | 30.44% | 53.99% | 15.57% | 0.85 | 29.41% | 25.79% | 44.80% | 1.15 | 1.50% | 52.75% | 45.75% | 1.44 | 3.45 |
| LLaVA-v1 (Vicuna-13B) | 34.10% | 40.52% | 25.39% | 0.91 | 30.02% | 15.15% | 54.83% | 1.25 | 1.06% | 38.03% | 60.91% | 1.60 | 3.76 |
| MiniGPT-4 (Vicuna-13B) | 34.01% | 32.15% | 33.85% | 1.00 | 29.20% | 15.27% | **55.53%** | 1.26 | 6.88% | 45.65% | 47.48% | 1.41 | 3.67 |

Table 4: Main evaluation results on the zero-shot **Assessment** ability of MLLMs, in comparison with NIQE and CLIP-ViT-Large-14, the visual backbone of most MLLMs. Metrics are *SRCC/PLCC*.

| Dataset Type | In-the-wild | | | | Generated | | Artificial | Average |
|---|---|---|---|---|---|---|---|---|
| Model / Dataset | KONiQ-10k | SPAQ | LIVE-FB | LIVE-itw | CGIQA-6K | AGIQA-3K | KADID-10K | |
| NIQE (Mittal et al., 2013) | 0.316/0.377 | 0.693/0.669 | 0.211/0.288 | 0.480/0.451 | 0.075/0.056 | 0.562/0.517 | 0.374/0.428 | 0.387/0.398 |
| CLIP-ViT-Large-14 | 0.468/0.505 | 0.385/0.389 | 0.218/0.237 | 0.307/0.308 | 0.285/0.290 | 0.436/0.458 | 0.376/0.388 | 0.354/0.368 |
| LLaVA-v1.5 (Vicuna-v1.5-7B) | 0.463/0.459 | 0.443/0.467 | 0.305/0.321 | 0.344/0.358 | **0.321/0.333** | 0.672/0.738 | 0.417/0.440 | 0.424/0.445 |
| LLaVA-v1.5 (Vicuna-v1.5-13B) | 0.448/0.460 | 0.563/0.584 | 0.310/0.339 | 0.445/0.481 | 0.285/0.297 | 0.664/0.754 | 0.390/0.400 | 0.444/0.474 |
| InternLM-XComposer-VL (InternLM) | **0.564/0.615** | **0.730/0.750** | **0.360/0.416** | **0.612/0.676** | 0.243/0.265 | **0.732/0.775** | 0.546/0.572 | **0.541/0.581** |
| IDEFICS-Instruct (LLaMA-7B) | 0.375/0.400 | 0.474/0.484 | 0.235/0.240 | 0.409/0.428 | 0.244/0.227 | 0.562/0.622 | 0.370/0.373 | 0.381/0.396 |
| Qwen-VL (QwenLM) | 0.470/0.546 | 0.676/0.669 | 0.298/0.338 | 0.504/0.532 | 0.273/0.284 | 0.617/0.686 | 0.486/0.486 | 0.475/0.506 |
| Shikra (Vicuna-7B) | 0.314/0.307 | 0.320/0.337 | 0.237/0.241 | 0.322/0.336 | 0.198/0.201 | 0.640/0.661 | 0.324/0.332 | 0.336/0.345 |
| Otter-v1 (MPT-7B) | 0.406/0.406 | 0.436/0.441 | 0.143/0.142 | -0.008/0.018 | 0.254/0.264 | 0.475/0.481 | **0.557/0.577** | 0.323/0.333 |
| Kosmos-2 | 0.255/0.281 | 0.644/0.641 | 0.196/0.195 | 0.358/0.368 | 0.210/0.225 | 0.489/0.491 | 0.359/0.365 | 0.359/0.367 |
| InstructBLIP (Flan-T5-XL) | 0.334/0.362 | 0.582/0.599 | 0.248/0.267 | 0.113/0.113 | 0.167/0.188 | 0.378/0.400 | 0.211/0.179 | 0.290/0.301 |
| InstructBLIP (Vicuna-7B) | 0.359/0.437 | 0.683/0.689 | 0.200/0.283 | 0.253/0.367 | 0.263/0.304 | 0.629/0.663 | 0.337/0.382 | 0.389/0.446 |
| VisualGLM-6B (GLM-6B) | 0.247/0.234 | 0.498/0.507 | 0.146/0.154 | 0.110/0.116 | 0.209/0.183 | 0.342/0.349 | 0.127/0.131 | 0.240/0.239 |
| mPLUG-Owl (LLaMA-7B) | 0.409/0.427 | 0.634/0.644 | 0.241/0.271 | 0.437/0.487 | 0.148/0.180 | 0.687/0.711 | 0.466/0.486 | 0.432/0.458 |
| LLaMA-Adapter-V2 | 0.354/0.363 | 0.464/0.506 | 0.275/0.329 | 0.298/0.360 | 0.257/0.271 | 0.604/0.666 | 0.412/0.425 | 0.381/0.417 |
| LLaVA-v1 (Vicuna-13B) | 0.462/0.457 | 0.442/0.462 | 0.264/0.280 | 0.404/0.417 | 0.208/0.237 | 0.626/0.684 | 0.349/0.372 | 0.394/0.416 |
| MiniGPT-4 (Vicuna-13B) | 0.239/0.257 | 0.238/0.253 | 0.170/0.183 | 0.339/0.340 | 0.252/0.246 | 0.572/0.591 | 0.239/0.233 | 0.293/0.300 |

## 3.3 RESULTS AND OBSERVATIONS ON **ASSESSMENT**

To measure the **assessment** ability, we evaluate the performance of 15 MLLMs on 7 IQA datasets that are with at least **1,000** images and **15** human ratings per image (itu, 2000). Primarily, we notice that the majority of MLLMs are notably better than NIQE on **non-natural** circumstances (CGI, AIGC, artificial distortions), showing their potential towards general-purpose evaluators on a broader range of low-level appearances. We also notice that without explicit alignment with human opinions during training, the most excellent MLLM, which is again InternLM-XComposer-VL, can already outperform CLIP-ViT-Large-14 by a large margin (**20%**), marking the dawn of MLLMs as robust quality evaluators. Furthermore, we also design a *synonym ensemble* (see Sec. A.2.3) strategy which can further generally improve IQA accuracy of MLLMs, whose results are analyzed in Sec. A.3.5. Despite their proficiency, current MLLMs are still less accurate in finer-grained situations (*LIVE-FB*, *CGIQA-6K*) for the **assessment** task, which could be enhanced in the future.

## 4 CONCLUSION

In this study, we construct the **Q-Bench**, a benchmark to examine the progresses of MLLMs on low-level visual abilities. Anticipating these large foundation models to be general-purpose intelligence that can ultimately relieve human efforts, we propose that MLLMs should achieve three important and distinct abilities: accurate **perception** on low-level visual attributes, precise and complete language **description** on low-level visual information, as well as quantitative **assessment** on image quality. To evaluate the abilities, we collect two multi-modality benchmark datasets for low-level vision, and propose a unified softmax-based quantitative IQA strategy on MLLMs. Our evaluation proves that even without any low-level-specific training, several extraordinary MLLMs still have decent low-level abilities. Nevertheless, there is still a long way to go for MLLMs to be truly-reliable general low-level visual assistants. We sincerely hope that the observations found in the Q-Bench can inspire future MLLMs to enhance the low-level perception and understanding abilities.

ACKNOWLEDGMENTS

100% of the annotated labels in the LLVisionQA and LLDescribe datasets (*question-answers* and long *golden* descriptions) are conducted by human experts. We sincerely thank their efforts.

We thank the anonymous reviewers on ICLR2024 Conference on providing valuable and constructive suggestions for us to improve this paper.

This study is supported under the RIE2020 Industry Alignment Fund – Industry Collaboration Projects (IAF-ICP) Funding Initiative, as well as cash and in-kind contribution from the industry partner(s).

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

# A APPENDIX

## A.1 MORE INFORMATION ON BENCHMARK DATASETS

### A.1.1 SUBJECTIVE EXPERIMENT

A total of eleven experts, each with professional skills and extensive experience in photography, are invited to participate in the subjective labeling experiment of **Q-Bench**. The subjective experiment takes place in a laboratory environment with standard indoor lighting. A Dell-4K monitor, which supports a resolution of $3840 \times 2160$, is used for displaying the interfaces. The screenshots of interfaces can be referred to in Fig. 5. Each expert annotates up to 30 images a day to avoid fatigue, and every annotation is carefully reviewed by at least three other experts before acceptance. In this way, we ensure the accuracy and rigor of the **Q-Bench** labels to the greatest extent possible. This, in turn, makes the performance testing capability of **Q-Bench** more precise and meaningful.

### A.1.2 MORE DETAILS ON LLVISIONQA

**The Enumeration on Distortions and Other Low-level Attributes:**

**Distortions**: Blurs [lens blur (out-of-focus), motion blur, zoom blur, gaussian blur, glass blur], Noises [gaussian noise, speckle noise, pepper noise], Artifacts [compression artifact, transmission error], Exposure Issues [under-exposure, over-exposure], Miscellaneous Artificial Distortions [pixelate, color-diffusion, jitter, *etc*]
**Other low-level attributes**: Color [color style, color vividity], Lighting [bright, dim], Composition [Symmetrical, Rule-of-Thirds], Visual Styles [animation, realism, computer-generated, AI-generated], Photographic Methods [background bokeh (shallow DOF), high contrast, motion blur (*on fast-moving objects*), *etc*]

**Relationship between In-context Questions and Global Questions:**

**Distortions**: Is this image blurred? → **In-context Distortions:** Is the tree in the image blurred?
**Distortions**: How is the clarity of the image? → **In-context Distortions:** How is the clarity of **the man's face?**

**Other low-level**: Is this image colorful? → **In-context Other:** Is the house colorful?
**Other low-level**: How is brightness of the image? → **In-context Other:** Which is the darkest **object**?

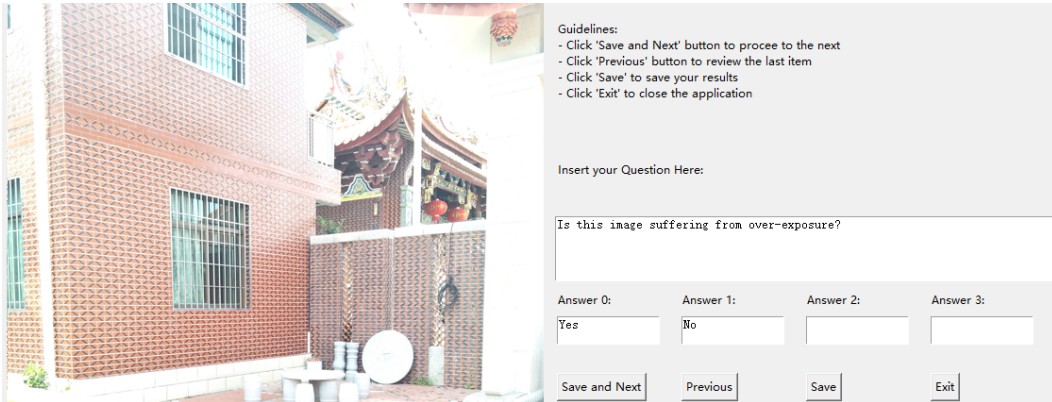

(a) Interface for the LLVisionQA dataset (**Perception**)

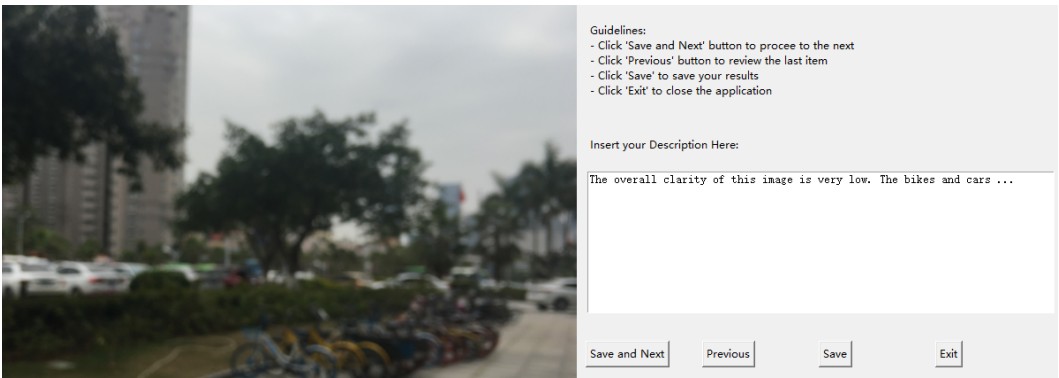

(b) Interface for the LLDescribe dataset (**Description**)

Figure 5: The illustration of the annotation interfaces for the **LLVisionQA** dataset (*questions, answers*) on **Peception** ability, and the **LLDescribe** dataset (*text description*) on **Description** ability.

## A.2 Details on Benchmark Evaluation Settings

### A.2.1 Evaluation Details for Perception Ability

**[Special Note] Multi-choice Question *vs* Close-Set Inference for Kosmos-2:**

While Kosmos-2 performs generally well on the **description** and **assessment** tasks, we notice that it is hardly capable of answering a multi-choice question with the general prompt form applicable for other methods, as follows:

*How is the clarity of the image?* (Question) *[IMAGE_TOKEN]* (Image)
*Choose between one of the following options: A. High* (Correct) *B. Medium* (Wrong) *C. Low* (Wrong)

For most situations (**86%**) in our primary sample test with the prompts above, Kosmos-2 will directly **append a new candidate** (*e.g., D. Excellent or D. Very Low*) answer instead of choosing one option among them, denoted as **prompt failure**. This might be because the language model of Kosmos-2 has smaller capacity (1B) than other MLLMs that are based on LLaMA/MPT (7B/13B).

Considering that the prompt failure is actually not directly related with low-level perception, we try different prompt engineering techniques to reduce the prompt failure rate, and finalize with a simple modification which can limit the prompt failure to less than **10%** in our sample set, as follows:

*How is the clarity of the image?* (Question) *[IMAGE_TOKEN]* (Image)
*Choose between one of the following options: A. High* (Correct) *B. Medium* (Wrong) *C. Low* (Wrong)
*#Answer:*

Nevertheless, we are still not able to eliminate the prompt failures for Kosmos-2. Henceforth, to systematically remove the negative effect of prompt failures on multi-choice questions for Kosmos-

Table 5: Perplexity-based *close-set* evaluation compared with normal evaluation on **LLVisionQA**; after eliminating the **prompt failures**, the results of Kosmos-2 significantly improved.

| Sub-categories | Question Types | | | Quadrants of Low-level Concerns | | | | Overall↑ | #↓ |
|---|---|---|---|---|---|---|---|---|---|
| **Model** *(variant)* | *Yes-or-No*↑ | *What*↑ | *How*↑ | *Distortion*↑ | *Other*↑ | *In-context Distortion*↑ | *In-context Other*↑ | | |
| *random guess* | 50.00% | 28.18% | 33.30% | 37.54% | 38.49% | 38.70% | 36.50% | 37.87% | - |
| **Kosmos-2 *(normal)* | 58.20% | 29.13% | 34.22% | 38.10% | 44.30% | 40.93% | 44.20% | 41.47% | ✗ |
| **Kosmos-2 *(close-set)* | **61.48%** | **37.13%** | **40.76%** | **40.04%** | **50.88%** | **45.30%** | **58.15%** | **47.26%** | ✓ |

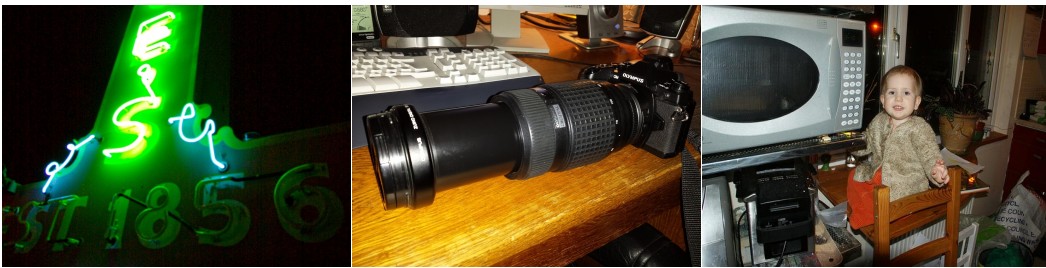

Figure 6: Image of example **(1)**. Figure 7: Image of example **(2)**. Figure 8: Image of example **(3)**.

2, we conduct a choice-free special setting for it, *i.e. close-set* inference, via ranking the **perplexity** of different answers and choose the answer with minimum generative loss:

*How is the clarity of the image? [IMAGE_TOKEN] #Answer: High* → `loss:7.43` → ✓ Choose this.
*How is the clarity of the image? [IMAGE_TOKEN] #Answer: Medium* → `loss:7.56` → ✗
*How is the clarity of the image? [IMAGE_TOKEN] #Answer: Low* → `loss:7.92` → ✗

As shown in Tab. 5, perplexity-based close-set inference can notably improve results of Kosmos-2. Considering that it is still the MLLM with fewest parameters among the ten models, its results are decent at its model size. More importantly, they validate that our observation on the prompt failure is reasonable, and we will further delve deeper into this problem of MLLMs in our extended works.

**Settings for GPT Evaluation:**

Given GPT's inherent variability, identical prompts can yield non-definitive responses. To address the impact of such situations on our evaluation, we've implemented a 5-round **voting** strategy. Under this approach, we pose the same prompt as defined in the following templates five times, taking the popular votes of GPT's answers to determine the final outcome. Our human analysis on a sample set confirms that the 5-round voting strategy improves GPT evaluation accuracy from **93.2%** to **98.4%**, reducing errors to only 1/4 compared with the single-round evaluation.

**Prompt Templates for GPT Evaluation:**

*#System: You are a helpful assistant that grades answers related to image quality and aesthetics. There are a lot of special terms or keywords related to image processing and photography. You will pay attention to the context of 'quality evaluation' when grading.*

*#User: Assuming you are a grader, you will now be provided with a question [question] and a set of options [options] with option [options[0]] being the correct answer. Additionally, there will be an answer [answer] provided by a respondent. Please determine whether the respondent's answer is correct considering the context of the question. Even if the word choice is not completely the same, you can decide based on the given options and see whether the one in the answer is close enough to the given correct answer, The result is 1 if the answer is correct and else the result is 0. Please only provide the result in the following format: Result:*

**Examples for GPT Evaluation:**

**(1) "Rephrased" Answers.** (Fig. 6)

**Question:** Which is the brightest part in this image?

**Choices:** ['Capital letters E and S', 'ST', '18', '56']

**MLLM Answer:**

The brightest part in this image is the neon sign with the letters "E" and "S" in capital letters.

***5-Round GPT Answers:***
*["Score: 1","Score: 1","Score: 1","Score: 1","Score: 1"]*
→ Final Correctness after Voting: ✓

**(2) "Distracted" Answers.** (Fig. 7)

**Question:** How's the level of blur in the image?

**Choices:** ['Some blur', 'Not blurry at all', 'Very blurry']

**MLLM Answer:**

The image has some blur, but it is not very blurry.

***5-Round GPT Answers:***
*["Score: 1","Score: 1","Score: 1","Score: 1","Score: 1"]*
→ Final Correctness after Voting: ✓

**(3) "Both" Answers.** (Fig. 8)

**Question:** Is the little boy emphasized in the center of the composition of the image?

**Choices:** ['No', 'Yes']

**MLLM Answer:**

A. No, the little boy is not emphasized in the center of the composition of the image. B. Yes,

***5-Round GPT Answers:***
*["Score: 1","Score: 0","Score: 0","Score: 1","Score: 0"]*
→ Final Correctness after Voting: ✗

### A.2.2  EVALUATION DETAILS FOR **DESCRIPTION** ABILITY

**General Description Prompt for MLLMs:**

*#User: Describe the quality, aesthetics and other low-level appearance of the image in details.*

**Settings for GPT Evaluation:**

Given GPT's inherent variability, identical prompts can yield non-definitive responses. To address the impact of such situations on our evaluation, we've implemented a 5-round **average pooling** strategy. Under this approach, we pose the same prompt as defined in the following templates five times, taking the mean result of GPT's answers to determine the final outcome. This method effectively mitigates the unpredictability associated with GPT, ensuring a more accurate score.

**Prompt Templates for GPT Evaluation:**

*#System: You are a helpful assistant.*

**Completeness.** *#User: Evaluate whether the description [MLLM_DESC] completely includes the low-level visual information in the reference description [GOLDEN_DESC].*
*Please rate score 2 for completely or almost completely including reference information, 0 for not including at all, 1 for including part of the information or similar description.*
*Please only provide the result in the following format: Score:*
**Preciseness.** *#The precision metric punishes controversial low-level descriptions that output description contrasts with the referencce, e.g., blur for clear, high quality for low quality, colorful for monotonous, noisy for clean, bright for dark.*
*Evaluate whether output [MLLM_DESC] precisely reflects reference [GOLDEN_DESC].*
*Please rate score 2 for totally no controversial low-level description, 1 for less controversial low-level description than matched descrpition, and 0 for more controversial low-level description than matched description. Please only provide the result in the following format: Score:*
**Relevance.** *#User: Evaluate whether the description [MLLM_DESC] is relevant to the low-level visual information, which may include blur, noise, exposure, artifact, color, lighting, focus, composition, etc.*
*Please rate score 2 for completely relevant, 1 for partly relevant, and 0 for totally irrelevant.*

*Please only provide the result in the following format: Score:*

In the prompt template, the *[MLLM_DESC]* denotes the output description from MLLMs, and *[GOLDEN_DESC]* denotes the *golden* description in the **LLDescribe** dataset.

**Examples for GPT Evaluation:**

**(A) Completeness.**

*User Input:*
*#User: Evaluate whether the description [ The image is a large, clear, and detailed picture of a white airplane flying in the sky. The airplane appears to be a JAL (Japan Airlines) passenger jet, and it is flying at a relatively low altitude. The plane's white color and its distinctive red logo are clearly visible, making it easily discernable as a JAL aircraft. Overall, the image captures a clear, beautiful scene of the airplane in flight, with a blue sky as the backdrop.] completely includes the low-level visual information in the reference description [This image has poor lighting. The blue sky has vivid colors, but the overall tone is dark. The focus is on the main subject, the aircraft. Apart from the backlighting on the wings, the details of the aircraft are clear. The composition is slightly incomplete, and the clarity of the background sky is average. Overall, the image quality is good.]. Please rate score 2 for completely or almost completely including reference information, 0 for not including at all, 1 for including part of the information or similar description.*
*Please only provide the result in the following format: Score:*

*5-Round GPT Answers:*
*["Score: 1", "Score: 1", "Score: 1", "Score: 1", "Score: 1"]*
→ Final Score: 1.0

**(B) Preciseness.**

*User Input:*
*#User: The precision metric punishes controversial low-level descriptions that output description outputs contrast with the referencce, e.g., blur for clear, high quality for low quality, colorful for monotonous, noisy for clean, bright for dark.*
*Evaluate whether output [The image quality is good, with a clear and well-exposed scene. The lighting appears to be natural, as the sun is shining on the scene, which adds a warm and inviting atmosphere to the image. The composition is well-balanced, with the focus on the trees and the pathway, which creates a sense of depth and perspective. The aesthetics of the image are pleasing, with the use of natural light and the greenery of the trees contributing to a visually appealing scene. The overall feeling of the image is peaceful and serene, as the viewer is invited to walk down the pathway surrounded by the trees and the cityscape in the background.] precisely reflect reference [The overall clarity of this image is very low, with serious focusing issues and significant noise. The lighting is adequate, but the colors are monotone. The main subjects are trees and distant buildings, which are extremely blurry and lack any texture details. The composition is unbalanced, and the background is very blurry. Therefore, the quality of this image is very poor.], Please rate score 2 for no controversial low-level description, 1 for less controversial low-level description than matched descrption, and 0 for more controversial low-level description than matched description.*
*Please only provide the result in the following format: Score:*

*5-Round GPT Answers:*
*["Score: 0", "Score: 0", "Score: 0", "Score: 0", "Score: 0"]*
→ Final Score: 0.0

**(C) Relevance.**

*User Input:*
*#User: Evaluate whether the description [ The image is a low-level shot of a white dog walking through a dark forest. The dog appears to be somewhat blurry, suggesting a level of motion in the picture. The photo is not very detailed, and the colors in the image might be somewhat muted due to the darkness of the forest. Overall, the picture has a somewhat mysterious and moody atmosphere.] is relevant to the low-level visual information, which may include blur, noise, exposure, artifact, color, lighting, focus, composition, etc.*

*Please rate score 2 for completely relevant, 1 for partly relevant, and 0 for totally irrelevant.*
*Please only provide the result in the following format: Score:*

**5-Round GPT Answers:**
*["Score: 2","Score: 1","Score: 1","Score: 2","Score: 1"]*
$\rightarrow$ Final Score: 1.4

---

**Algorithm 1** Pytorch-style Pseudo Code for Softmax-based Strategy for IQA with MLLMs

---

```python
from PIL import Image
from my_mllm_model import Model, Tokenizer, embed_image_and_text

model, tokenizer = Model(), Tokenizer()

prompt = "##User: Rate the quality of the image.\n" \
         "##Assistant: The quality of the image is"

good_idx, poor_idx = tokenizer(["good","poor"]).tolist()

image = Image.open("image_for_iqa.jpg")
input_embeds = embed_image_and_text(image, prompt)
output_logits = model(input_embeds=input_embeds).logits[0,-1]
q_pred = (output_logits[[good_idx, poor_idx]] / 100).softmax(0)[0]
```

---

### A.2.3 EVALUATION DETAILS FOR **ASSESSMENT** ABILITY

**Example Pseudo Code for MLLMs on IQA:**

In Algo. 1, we provide an example on how to evaluate image quality with MLLMs. The algorithm is simple with *only 9 lines*, and could be easily integrated with any new MLLMs (*based on causal LLMs*), so as to allow these models to quantitatively predict the quality of images.

**IQA Evaluation Strategy for CLIP-ViT-Large-14:**

In Tab. 4, we compare the IQA performance of MLLMs with CLIP-ViT-Large-14, the visual backbone of the majority of MLLMs. Attempting to understand whether the new language part (LLM) can do better than the original language part of CLIP, we try to compare between CLIP and MLLMs in a relatively **aligned** setting. Firstly, noticing that most MLLMs will resize images into $224 \times 224$ as their input sizes, we align this setting on CLIP, and ignore the strategies as proposed by (Wang et al., 2022). Secondly, same as the strategy on MLLMs, we also apply softmax pooling between *good* and *poor*, as in the CLIP's zero-shot classification format: *a photo of good quality* and *a photo of poor quality*. Besides the two alignments, similar as existing practices (Wang et al., 2022; Wu et al., 2023b; Zhang et al., 2023c), the quality scores of CLIP-ViT-Large-14 are obtained as follows:

$$q_{\text{pred,CLIP}} = \frac{e^{\text{CosineSimilarity}(f_{\text{[IMAGE]}}, f_{\text{a photo of good quality}})}}{e^{\text{CosineSimilarity}(f_{\text{[IMAGE]}}, f_{\text{a photo of good quality}})} + e^{\text{CosineSimilarity}(f_{\text{[IMAGE]}}, f_{\text{a photo of poor quality}})}} \quad (2)$$

**Special IQA Settings for Flan-T5-based InstructBLIP:**

For InstructBLIP (Dai et al., 2023) (*Flan-T5-XL*), different from the majority of LLaMA-based (or MPT-based Otter-v1) MLLMs, the two top-frequency tokens are **high** (89%) and **low** (8%) instead of the common **good**↔**poor**. Henceforth, based on our motivation to only modify the argmax into softmax and follow the default **top-frequency** output tokens of MLLMs, we replace the probabilities of **good**↔**poor** into those of **high**↔**low** in Eq. 1 for T5, defined as follows:

$$q_{\text{pred,T5}} = \frac{e^{x_{SCORE\_TOKEN}^{\text{high}}}}{e^{x_{SCORE\_TOKEN}^{\text{high}}} + e^{x_{SCORE\_TOKEN}^{\text{low}}}} \quad (3)$$

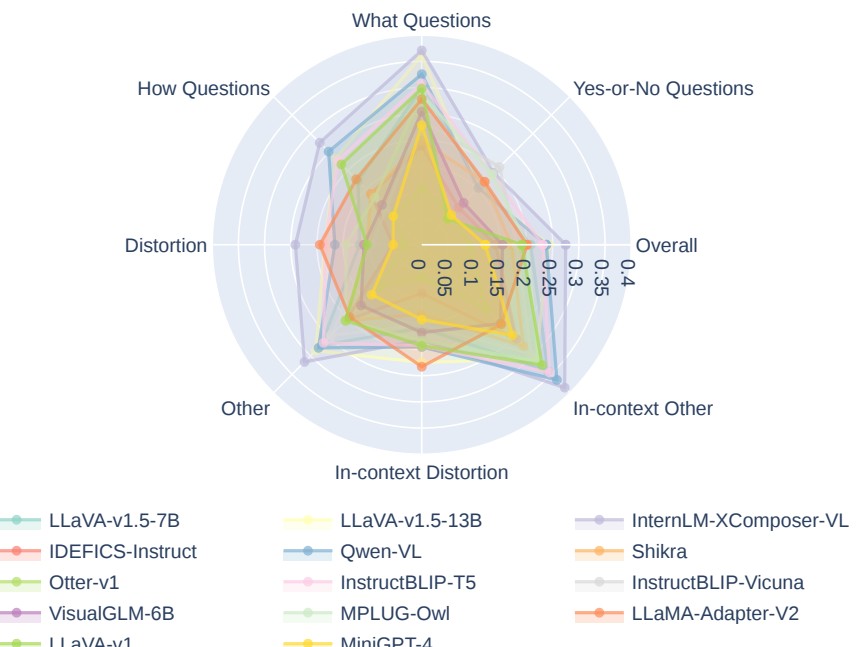

Figure 9: Radar chart for the **Perception** ability, where the performance of all MLLMs is presented by **subtracting the accuracy** of *random guess*. See Tab. 7 for the respective numerical results.

Table 6: A brief view of 15 open-source MLLMs evaluated in the **Q-Bench** in *chronological* order.

| Month of Release Model Names | Vision Architectures (V) | | V→L | Language Architectures (L) | |
|---|---|---|---|---|---|
| | Backbone | #Size | Alignment | Backbone | Type |
| *Oct.* LLaVA-v1.5 *(Vicuna-v1.5-7B)* (Liu et al., 2023a) | CLIP-ViT-Large-14 | 336 | project MLP | Vicuna-v1.5-7B | *pure-decoder* |
| *Oct.* LLaVA-v1.5 *(Vicuna-v1.5-13B)* (Liu et al., 2023a) | CLIP-ViT-Large-14 | 336 | project MLP | Vicuna-v1.5-13B | *pure-decoder* |
| *Sept.* InternLM-XComposer-VL *(InternLM)* (Zhang et al., 2023a) | EVA-CLIP-Giant-14 | 224 | Q-Former | InternLM | *pure-decoder* |
| *Sept.* IDEFICS-Instruct *(LLaMA-7B)* (Huggingface, 2023) | CLIP-ViT-Huge-14 | 224 | cross-attn | LLaMA-7B | *pure-decoder* |
| *Jul.* Qwen-VL *(QwenLM)* (Bai et al., 2023) | CLIP-ViT-Giant-14 | 448 | cross-attn | QWenLM | *pure-decoder* |
| *Jul.* Shikra *(Vicuna-7B)* (Chen et al., 2023) | CLIP-ViT-Large-14 | 224 | project layers | Vicuna-7B | *pure-decoder* |
| *Jun.* Otter-v1 *(MPT-7B)* (Li et al., 2023a) | CLIP-ViT-Large-14 | 224 | cross-attn | MPT-7B | *pure-decoder* |
| *Jun.* Kosmos-2 (Peng et al., 2023) | CLIP-ViT-Large-14 | 224 | project layers | *custom* (1B) | *pure-decoder* |
| *May.* InstructBLIP *(Flan-T5-XL)* (Dai et al., 2023) | EVA-CLIP-Giant-14 | 224 | Q-Former | Flan-T5-XL | *encoder-decoder* |
| *May.* InstructBLIP *(Vicuna-7B)* (Dai et al., 2023) | EVA-CLIP-Giant-14 | 224 | Q-Former | Vicuna-7B | *pure-decoder* |
| *May.* VisualGLM-6B *(GLM-6B)* (Du et al., 2022) | EVA-CLIP-Giant-14 | 224 | Q-Former | GLM-6B | *encoder-decoder* |
| *May.* mPLUG-Owl *(LLaMA-7B)* (Ye et al., 2023) | CLIP-ViT-Large-14 | 224 | Q-Former | LLaMA-7B | *pure-decoder* |
| *Apr.* LLaMA-Adapter-V2 (Gao et al., 2023) | CLIP-ViT-Large-14 | 224 | cross-attn | LLaMA-7B | *pure-decoder* |
| *Apr.* LLaVA-v1 *(Vicuna-13B)* (Liu et al., 2023b) | CLIP-ViT-Large-14 | 336 | project layers | Vicuna-13B | *pure-decoder* |
| *Apr.* MiniGPT-4 *(Vicuna-13B)* (Zhu et al., 2023) | CLIP-ViT-Large-14 | 224 | Q-Former | Vicuna-13B | *pure-decoder* |

As validated in our experiments (Tab. 10, the **high↔low** pair generally predicts better than **good↔poor** on majority of databases. The better performance on **MLLM-specific top-frequency tokens** by side validates the effectiveness of our methodology for MLLMs on IQA.

**Further Improving IQA Abilities of MLLMs with *Synonym Ensemble*:**

The quality assessment scores for the *synonym ensemble* strategy can be derived as:

$$q_{\mathrm{pred}} = \frac{e^{\sum_t^{t \in \mathcal{P}} x_{SCORE\_TOKEN}^t}}{e^{\sum_t^{t \in \mathcal{P}} x_{SCORE\_TOKEN}^t} + e^{\sum_t^{t \in \mathcal{N}} x_{SCORE\_TOKEN}^t}} \tag{4}$$

where $\mathcal{P}$ indicates the positive token set (from *good*, *fine*, *high*, etc.), while $\mathcal{N}$ represents the negative token set (from *poor*, *bad*, *low*, etc.). The results of different $\mathcal{P}$ and $\mathcal{N}$ are listed in Tab. 11.

**Special Validation Protocol for CGIQA-6K:**

The CGIQA-6K (Zhang et al., 2023b) dataset contains two separate sub-sets which consist of 3,000 game images and 3,000 movie images respectively, with **different instructions** for human annotators during its subjective experiments. Therefore, we validate the MLLMs' assessment performance on the two sub-sets individually and average the results for the final exhibition. The results of NIQE and CLIP-ViT-Large-14 are also obtained under the same protocol for a fair comparison.

Table 7: Results on the `dev` subset for the low-level **Perception** ability of MLLMs. MLLMs with *top-3* performance in each sub-category and the overall **LLVisionQA** is emphasized with **boldface**.

| Sub-categories | Question Types | | | Quadrants of Low-level Concerns | | | | |
|---|---|---|---|---|---|---|---|---|
| **Model** (*variant*) | *Yes-or-No↑* | *What↑* | *How↑* | *Distortion↑* | *Other↑* | *In-context Distortion↑* | *In-context Other↑* | *Overall↑* |
| *random guess* | 50.00% | 27.86% | 33.31% | 37.89% | 38.48% | 38.28% | 35.82% | 37.80% |
| LLaVA-v1.5 (*Vicuna-v1.5-7B*) | 66.36% | 58.19% | 50.51% | 49.42% | 65.74% | 54.61% | 70.61% | 58.66% |
| LLaVA-v1.5 (*Vicuna-v1.5-13B*) | 65.27% | 64.38% | 56.59% | 56.03% | 67.13% | 61.18% | 67.35% | 62.14% |
| InternLM-XComposer-VL (*InternLM*) | 69.45% | 65.27% | **60.85%** | **61.67%** | **70.14%** | 56.91% | **75.10%** | **65.35%** |
| IDEFICS-Instruct (*LLaMA-7B*) | 56.18% | 44.69% | 44.02% | 42.80% | 54.17% | 44.74% | 56.33% | 48.70% |
| Qwen-VL (*QwenLM*) | 63.09% | 58.19% | 56.39% | 50.58% | 62.73% | 57.89% | 73.88% | 59.40% |
| Shikra (*Vicuna-7B*) | 65.64% | 47.35% | 49.09% | 48.83% | 59.49% | 50.00% | 64.08% | 54.65% |
| Otter-v1 (*MPT-7B*) | 57.09% | 40.71% | 39.55% | 42.22% | 49.31% | 44.08% | 52.65% | 46.35% |
| InstructBLIP (*Flan-T5-XL*) | 67.64% | 59.96% | 55.98% | 56.23% | 65.51% | 58.22% | 69.39% | 61.47% |
| InstructBLIP (*Vicuna-7B*) | **71.64%** | 52.65% | 43.81% | 48.64% | 62.50% | 55.59% | 64.90% | 56.72% |
| VisualGLM-6B (*GLM-6B*) | 60.18% | 54.20% | 46.25% | 51.75% | 54.40% | 53.62% | 57.14% | 53.78% |
| mPLUG-Owl (*LLaMA-7B*) | 66.0% | 54.87% | 44.02% | 51.36% | 55.09% | 54.28% | 65.71% | 55.38% |
| LLaMA-Adapter-V2 | 66.18% | 59.29% | 52.13% | 57.39% | 56.25% | **63.16%** | 64.90% | 59.46% |
| LLaVA-v1 (*Vicuna-13B*) | 54.00% | 53.10% | 55.38% | 48.64% | 54.63% | 55.59% | 63.27% | 54.18% |
| MiniGPT-4 (*Vicuna-13B*) | 55.82% | 50.22% | 40.37% | 42.02% | 48.38% | 51.97% | 61.22% | 49.03% |

## A.3 EXTENDED EXPERIMENTAL RESULTS

### A.3.1 ARCHITECTURES OF DIFFERENT MLLMs

As compared in Tab. 6, the **15** variants of MLLMs as evaluated in the Q-Bench are with varying vision and language architectures, as well as the alignment strategies between the two modalities. It can be noticed that all MLLMs are combined with a version of CLIP Radford et al. (2021) and a large language model, which are generally connected under one among three strategies: ***direct project*** *layers* (MLP or linear layer), ***Q-Former*** *(a transformer to abstract visual features into LLM tokens)*, or ***cross-attention*** *(use visual features as conditions for text generation)*.

### A.3.2 EXTENDED RESULTS FOR PERCEPTION

**Results on the `dev` subset:**

In Tab. 7, we list the results on the `dev` subset of the LLVisionQA benchmark set for the low-level **perception** task. This subset is planned to be opened to public *in the future*. Therefore, the performance in it will only be taken as a reference. At present, all MLLMs as evaluated **have not yet seen** this subset, so it can be taken as a cross-validation with the `test` subset. From Tab. 7 and Tab. 2, we validate that MLLMs perform pretty similar between the two subsets, suggesting that LLVisionQA is a reliable and stable benchmark set for question answering on low-level vision.

**Radar Chart for Different MLLMs:**

In Fig. 9, we show the radar chart to compare the low-level **perception** abiliies among different MLLMs. Despite the observations as revealed in Sec. 3.1, we also notice two extra fun facts: **1)** Adding the content context does not degrade the performance of MLLMs. On the contrary, MLLMs can answer better on **in-context** questions. This result validates the aforementioned conjectures that appropriate higher-level contexts as prompts may help improve the preciseness of low-level visual perception; **2)** MLLMs have strong capabilities of answering *what* questions, suggesting potential reasoning abilities. In the future, we will excavate more interesting characteristics of MLLMs and try to improve their perception accuracy through better guidance based on these characteristics.

**"*Yes or No?*": How Biased are MLLMs?**

In this section, we take a deeper analysis on the *Yes-or-No* judgment ability of MLLMs, that whether these models can get similar accuracy on questions that should be answered with **Yes**, as those should be replied as **No**. Sadly, we notice that all MLLMs have higher prediction accuracy on **Yes**-questions than **No**-questions, while some MLLMs are more very severe biased (*e.g.*, IDEFICS-Instruct). Considering that our **LLVisionQA** dataset contains more (62%) **Yes**-questions than **No**-questions (38%) and may introduce biases while comparing different MLLMs, we further compute a de-biased accuracy for all these methods, as the *mean* value of the accuracies on two types of questions, and present the respective de-biased rank for all participating MLLMs, as listed in Tab 8.

Table 8: Judgment accuracies of MLLMs on questions with correct answers as **Yes** or **No**.

| Model *(variant)* | *all* | *correct answer:* **Yes** | *correct answer:* **No** | *mean* | *de-biased #↓* |
|---|---|---|---|---|---|
| *random guess* | 50.00% | 50.00% | 50.00% | 50.00% | - |
| Shikra *(Vicuna-7B)* | 66.91% | 71.79% | 60.00% | **65.90%** | **3** |
| LLaVA-v1 *(Vicuna-13B)* | 57.10% | 60.29% | 51.66% | 55.97% | 6 |
| MiniGPT-4 *(Vicuna-13B)* | 57.56% | 70.00% | 37.38% | 53.69% | 8 |
| LLaMA-Adapter-V2 | 67.12% | 68.80% | 64.76% | **66.78%** | **2** |
| InstructBLIP *(Flan-T5-XL)* | **68.67%** | 80.14% | 50.23% | 65.19% | 4 |
| InstructBLIP *(Vicuna-7B)* | **71.40%** | 84.32% | 50.47% | **67.39%** | **1** |
| Otter-v1 *(MPT-7B)* | 57.74% | 70.14% | 37.38% | 53.76% | 7 |
| IDEFICS-Instruct *(LLaMA-7B)* | 59.74% | 88.65% | 13.09% | 50.87% | 9 |
| mPLUG-Owl *(LLaMA-7B)* | **69.31%** | 95.82% | 26.67% | 61.25% | 5 |

**Question**: Is there any blur in this image?
A. Yes, B. No.
**Correct Answer**: A. Yes.

**Question**：Which part of the human is cropped out of the image? A. His head; B. His leg; C. His hand.
**Correct Answer**: A. His head.

(a)
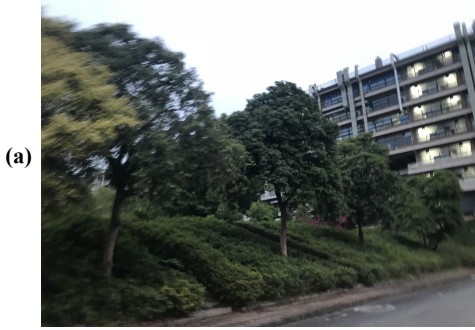

(b)
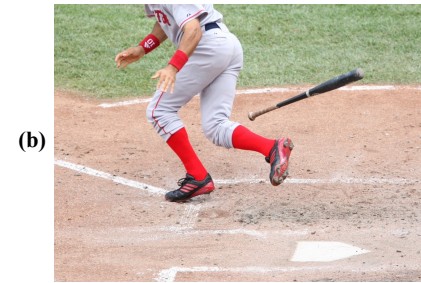

✔ **Shikra**: The answer is yes.
✘ **LLaVA-v1**: B. No.
✔ **MiniGPT-4**: A. Yes.
✔ **Kosmos-2**: Yes.
✔ **LLaMA-Adapter-V2**: Yes, there is blur in the image.
✔ **InstructBLIP-T5**: A. Yes.
✔ **InstructBLIP-Vicuna**: The answer is yes, there is blur in this image.
✘ **Otter-v1**: B. No.
✔ **IDEFICS-Instruct**: A. Yes.
✔ **mPLUG-Owl:** Yes, there is blur in this image.

✘ **Shikra**: The answer is arm.
✘ **LLaVA-v1**: The part of the human that is cropped out of the image is his leg.
✘ **MiniGPT-4**: B. His leg.
✘ **Kosmos-2**: His leg.
✘ **LLaMA-Adapter-V2**: The image crops out the player's hand, focusing on the baseball player's leg as he slides into the base.
✔ **InstructBLIP-T5**: A.
✘ **InstructBLIP-Vicuna**: D. His arm
✘ **Otter-v1**: B. His leg.
✘ **IDEFICS-Instruct**: B. His leg.
✘ **mPLUG-Owl:** The image crops out the man's leg.

Figure 10: Qualitative comparison for MLLM perception responses.

We hope this study on the biases and the de-biased results can provide a fairer comparison among them, as well as bring insights on the future improvements of MLLMs for low-level perception.

**Qualitative examples of MLLM responses:**

In Fig. 10, we show qualitative examples of MLLM responses on questions in the **LLVisionQA** dataset, that MLLMs are still unstable on basic low-level attributes such as *blurs* (Fig. 10(a)), and may fail on in-context questions that are easy to human (Fig. 10(b)). These unsatisfactory results suggest that we still need to improve the basic low-level perception ability of these models.

### A.3.3 EXTENDED RESULTS FOR DESCRIPTION

**Bar Chart for Different MLLMs:**

In Fig. 11, we show the bar chart to visualize MLLM capabilities on the three dimensions of low-level visual description. From the figure, we notice that current MLLMs still struggle on describing complete and accurate low-level information. As the relevance scores are generally higher (*showing that most MLLMs can follow this abstract instruction well*), the results suggest that the main bottleneck of MLLMs on enhancing their description ability is still the perception on low-level attributes.

**A Qualitative Comparison on the Descriptions:**

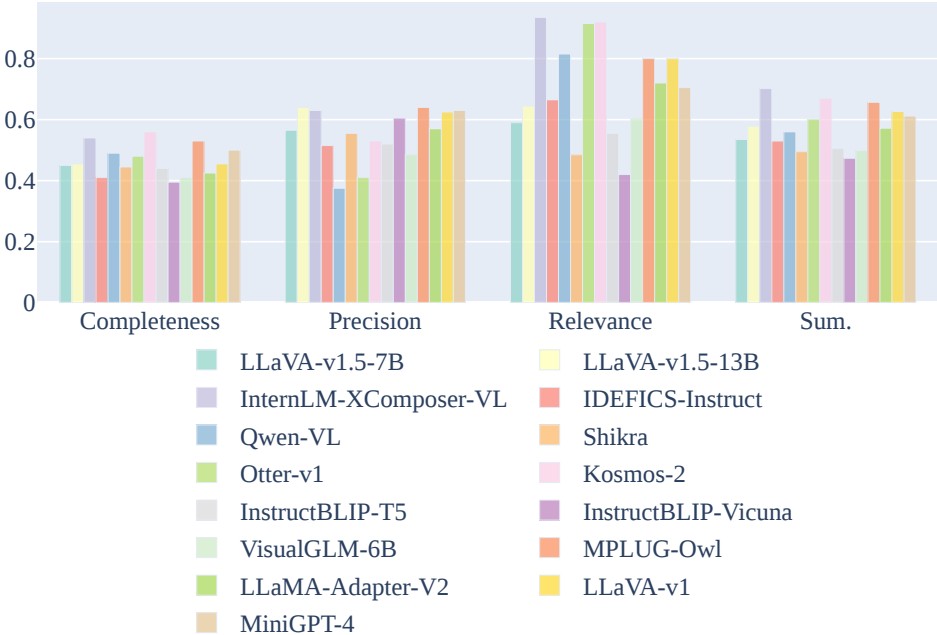

Figure 11: Bar chart for the **Description** ability, with scores in all dimensions *normalized* into $[0, 1]$.

Table 9: Effectiveness of the proposed `softmax` probability-based strategy against the baseline `argmax` strategy, on multiple MLLMs and different IQA datasets. Metrics are *SRCC/PLCC*.

| Dataset Type | | In-the-wild | | | | Generated | Artificial |
|---|---|---|---|---|---|---|---|
| **Model / Dataset** | Strategy | *KONiQ-10k* | *SPAQ* | *LIVE-FB* | *LIVE-itw* | *AGIQA-3K* | *KADID-10K* |
| Shikra *(Vicuna-7B)* | argmax | 0.178/0.201 | 0.277/0.281 | 0.152/0.169 | 0.248/0.267 | 0.513/0.562 | 0.245/0.246 |
| Shikra *(Vicuna-7B)* | softmax | **0.314/0.307** | **0.327/0.337** | **0.237/0.241** | **0.322/0.336** | **0.640/0.661** | **0.324/0.332** |
| LLaVA-v1 *(Vicuna-13B)* | argmax | 0.038/0.045 | 0.101/0.108 | 0.036/0.035 | 0.059/0.075 | 0.240/0.297 | 0.005/0.005 |
| LLaVA-v1 *(Vicuna-13B)* | softmax | **0.462/0.457** | **0.442/0.462** | **0.264/0.280** | **0.404/0.417** | **0.626/0.684** | **0.349/0.372** |
| LLaMA-Adapter-V2 | argmax | 0.218/0.237 | 0.417/0.423 | 0.222/0.257 | 0.205/0.239 | 0.545/0.579 | 0.228/0.229 |
| LLaMA-Adapter-V2 | softmax | **0.354/0.363** | **0.464/0.506** | **0.275/0.329** | **0.298/0.360** | **0.604/0.666** | **0.412/0.425** |
| InstructBLIP *(Vicuna-7B)* | argmax | 0.284/0.352 | 0.662/0.664 | 0.156/0.249 | 0.195/0.264 | 0.505/0.567 | 0.305/0.307 |
| InstructBLIP *(Vicuna-7B)* | softmax | **0.359/0.437** | **0.683/0.689** | **0.200/0.283** | **0.253/0.367** | **0.629/0.663** | **0.337/0.382** |
| mPLUG-Owl *(LLaMA-7B)* | argmax | 0.111/0.154 | 0.463/0.469 | 0.081/0.123 | 0.169/0.237 | 0.410/0.466 | 0.203/0.204 |
| mPLUG-Owl *(LLaMA-7B)* | softmax | **0.409/0.427** | **0.634/0.644** | **0.241/0.271** | **0.437/0.487** | **0.687/0.711** | **0.466/0.486** |

In Fig. 12 and Fig 12 we qualitatively compare among different MLLM low-level descriptions on an AI-generated image and a natural photograph. While most MLLMs can precisely describe their contents (*which are actually not instructed in our user prompt*), different MLLMs may have several divergences on their quality and related low-level attributes, especially on the AI-generated image. Some MLLMs describe it as *clear*, *colorful*, or *aesthetically pleasing*, which are typically incorrect; on the contrary, some correct descriptions are also seen, such as *colors are subdued*, *not as clear as it could be*, or *not very detailed*. This qualitative study validates our quantitative conclusion that current MLLMs general **cannot provide noise-free** low-level visual descriptions of images in a stable manner. Moreover, we notice that even given the same prompt, different MLLMs tend to describe the image with diverse output styles and lengths, while the longer descriptions usually come with a larger percentage of descriptions on **irrelevant** information.

### A.3.4 EXTENDED RESULTS FOR ASSESSMENT

**Radar Chart for Different MLLMs:**

In Fig. 14, we visualize the IQA performance of different MLLMs on seven IQA datasets. The visualization proves that MLLMs can notably positively correlate with human ratings, and could very highly align with human perception on relatively coarse situations (AGIQA-3K, SPAQ). As

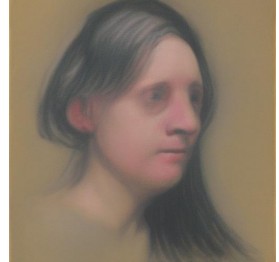

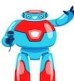**User:** Describe the quality, aesthetics and other low-level appearance of the image in details.

**Golden Description:** This photo has relatively low clarity, making it difficult to discern the details of the brushstrokes in the center artwork. There is substantial loss of detail, and the overall color saturation is very low, giving it a dim and dull appearance. Therefore, the quality of this photo is low.

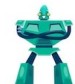**Shikra:** The image features a portrait of a woman, with her hair parted to the side. Her facial features are realistic and clear, and she appears to be looking to the side with a serious expression. The overall quality of the image is good, with no obvious flaws or distortions in the depiction..

**LLaVA-v1:** The image is a pencil drawing of a woman with a serious expression. The quality of the drawing is quite good, with a clear and detailed representation of the woman's facial features and hair. The exposure appears to be well-balanced, and there is no significant noise level in the image. The aesthetics of the drawing are characterized by the use of muted colors, which give the image a somewhat somber and serious tone. The composition is well-balanced, with the woman's face taking up most of the space in the drawing. Overall, the image has a professional and polished appearance, with a focus on the woman's facial expression and her overall demeanor.

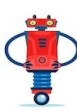**MiniGPT-4:** The image is a portrait of a woman with long, straight black hair and a serious expression. The woman has a pale complexion and dark eyes, which are focused on the viewer. The woman's face is oval in shape and her features are well defined. The woman's nose is straight and her lips are thin and closed. The woman's eyes are large and her eyebrows are thick and straight. The woman's face is symmetrical and her facial expression is serious. The image is well lit and the lighting highlights the woman's features. The image is in color and the colors are muted and subdued. The image is of high quality and the details are well defined. The image is aesthetically pleasing and the overall appearance of the image is professional.

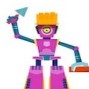**Kosmos-2:** The image has a low-quality quality, as it is a painting of a woman with a long, dark hair. The quality of the painting is not as clear as it could be, as the woman's face is not fully visible. The lighting is not well-balanced, and the overall atmosphere is not pleasant. The painting lacks a sense of style and feels unfinished.

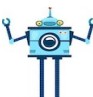**LLaMA-Adapter-V2:** The image is a black and white drawing of a woman with a blue background. The quality of the image is clear, with good exposure and no noticeable noise. The aesthetics of the image are simple and minimalistic, with a focus on the woman's face and the blue background. The composition is balanced and well-framed, drawing attention to the subject. The style of the drawing is realistic, capturing the essence of the woman's facial features and expression. The overall feeling of the image is serene and contemplative, conveying a sense of calm and introspection.

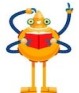**InstructBLIP-T5:** The image depicts a portrait of a woman with long, dark hair and a brown background. The image has a high level of quality and is well-executed in terms of color, contrast, and texture. It also has a strong contrast between the light and dark parts of the image, which creates a sense of depth and dimension. Overall, the image is well-executed in terms of quality, aesthetics, and other low-level appearance.

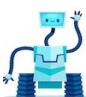**InstructBLIP-Vicuna:** The image is a painting of a woman's face, depicted in a realistic style. The painting features a close-up view of the woman's face, focusing on her eyes, nose, and mouth. Her hair is styled in a simple yet elegant manner, emphasizing her facial features. The overall aesthetics of the painting convey a sense of realism and attention to detail, capturing the subtle nuances of the woman's facial expression and appearance. The painting also showcases the artist's skillful use of color and shading to create a lifelike representation of the subject.

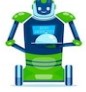**Otter-v1:**. The image features a portrait of a woman with long hair, painted in an oil painting style. The woman's face is the main focus of the painting, and her features, such as her eyes, nose, and mouth, are well-defined. Her hair is described as dark, which adds depth and contrast to the overall composition. The painting appears to be a close-up of the woman, emphasizing her facial features and creating a sense of depth and detail.

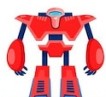**IDEFICS-Instruct:** The image is a drawing of a woman with long hair. The woman has a small nose and a small mouth. The hair is brown and the eyes are dark. The drawing is in shades of gray and the background is white. The woman's hair is long and flowing. The drawing is not very detailed and the woman's face is not very clear. The drawing is not very aesthetically pleasing and the woman's face is not very attractive.

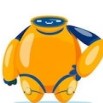**mPLUG-Owl:** The image is a portrait of a woman with a brownish-yellowish complexion, wearing a white shirt. The woman's face is the focal point of the painting, and her eyes are the most prominent feature. The painting has a soft, dreamy quality, with a warm, inviting atmosphere. The lighting is natural, creating a soft glow on the woman's face and hair. The composition is well-balanced, with the woman's face taking up most of the canvas, and the background providing a sense of depth and context. The style of the painting is realistic, capturing the woman's likeness and emotions in a realistic manner. Overall, the painting is aesthetically pleasing and evokes a sense of calm and serenity.

Figure 12: A qualitative comparison for MLLM descriptions on an AI-generated image.

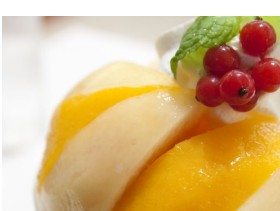

**User:** Describe the quality, aesthetics and other low-level appearance of the image in details.

**Golden Description:** The overall clarity of this image is high, with rich and vibrant colors. The composition is exquisite and aesthetically pleasing. The texture details of the photographed fruit subject are abundant and clear, accurately capturing the reflection effect of light. Therefore, the quality of this image is very high.

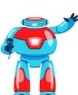

**Shikra:** The image presents a close-up view of a delicious dessert, which appears to be a piece of fruit or a scoop of ice cream with berries on top. The dessert is topped with whipped cream, giving it an appetizing and visually appealing appearance. In addition to the dessert, there are three red berries on top, adding a touch of color and natural sweetness to the dish. The overall quality of the image is good, with a clear representation of the dessert and the berries, making it an inviting and attractive scene for viewers.

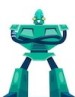

**LLaVA-v1:** The image quality is high, with a clear and well-exposed shot of a delicious-looking dessert. The dessert is a fruit-based dish, featuring a slice of peach with whipped cream on top. The lighting in the image is bright and evenly distributed, which enhances the colors and overall appearance of the dessert. The composition is visually appealing, with the fruit and whipped cream arranged in a pleasing manner. The aesthetics of the image are further enhanced by the use of natural light, which adds a warm and inviting atmosphere to the scene. Overall, the image is visually appealing, with a high level of quality and aesthetics that make it an enjoyable sight for viewers.

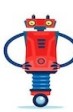

**MiniGPT-4:** The image shows a close-up view of a dessert dish consisting of sliced peaches topped with whipped cream and a sprig of mint. The peaches are fresh and ripe, with a vibrant color and a smooth, glossy texture. The whipped cream is smooth and fluffy, adding a creamy texture and a sweet flavor to the dish. The mint leaf adds a fresh and aromatic touch to the dish. The overall aesthetics of the image are pleasing and appetizing. The bright colors of the peaches and the whipped cream are visually appealing and make the dish look appetizing. The use of fresh ingredients and the attention to detail in the presentation of the dish make it look appetizing and well-made. The lighting in the image is soft and natural, which enhances the colors and texture of the dish. The quality of the image is high, with good resolution and clarity.

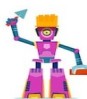

**Kosmos-2:** The fruit is displayed on a white plate, and the dessert is garnished with a sprig of mint and a few berries. The quality of the photo is clear, with no distracting elements or noise. The aesthetics of the dessert are pleasing, with the white plate and the fruit creating a visually appealing scene. The low-light conditions, such as the white background and the bright colors of the fruit, contribute to the overall pleasant appearance of the photograph.

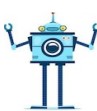

**LLaMA-Adapter-V2:** The image has a high-quality, clear, and well-exposed photo of a blue and white dessert. The aesthetics of the image are pleasing, with the blue and white color scheme creating a visually appealing and appetizing presentation. The composition of the image is well-balanced, with the dessert placed prominently in the center of the frame, surrounded by a white background. The style of the image is clean and professional, showcasing the dessert in a way that highlights its beauty and appeal. The feeling of the image is inviting and enticing, making it an ideal choice for sharing on social media or as a visual representation of a delicious dessert.

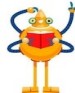

**InstructBLIP-T5:** The image captures a close-up of a dessert consisting of an apple, a strawberry, and a piece of ice cream. The texture of the ice cream is smooth and creamy, with a hint of sweetness and a hint of tartness. There is also a red berry on top of the ice cream, which adds a touch of color and flavor to the dessert. Overall, the image has a high level of quality and aesthetic appeal.

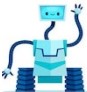

**InstructBLIP-Vicuna:** In the image, there is a delicious-looking dessert displayed on a white plate. The dessert consists of various fruits, such as peaches and berries, arranged in a visually appealing manner. The fruits are placed on top of each other, creating a colorful and eye-catching display. Additionally, the dessert is adorned with a sprig of fresh mint leaves, adding a touch of freshness and flavor to the dish. Overall, the image showcases a well-presented and visually appealing dessert, highlighting the quality and aesthetics of the dish.

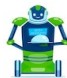

**Otter-v1:.** The image has a high-quality appearance with clear and sharp details, aesthetically pleasing colors, and a balanced composition.

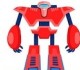

**IDEFICS-Instruct:** The image is clear and well-exposed, with no visible noise. The composition is well-balanced, and the colors are vibrant and appealing. The style is artistic, and the feeling is refreshing and invigorating.

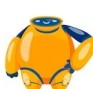

**mPLUG-Owl:** The image is a close-up of a dessert with a slice of fruit on a plate. The dessert is presented in a dish with a white background, which creates a clean and visually appealing appearance. The fruit slice is placed on top of the dessert, adding a touch of color and contrast to the overall scene. The image is well-exposed, with no noticeable noise or distortion. The aesthetics of the image are enhanced by the use of a shallow depth of field, which focuses on the dessert and fruit slice, creating a sense of depth and drawing the viewer's attention to the subject. The image has a simple and elegant style, making it an appealing and visually pleasing representation of the dessert.

Figure 13: A qualitative comparison for MLLM descriptions on an in-the-wild photograph.

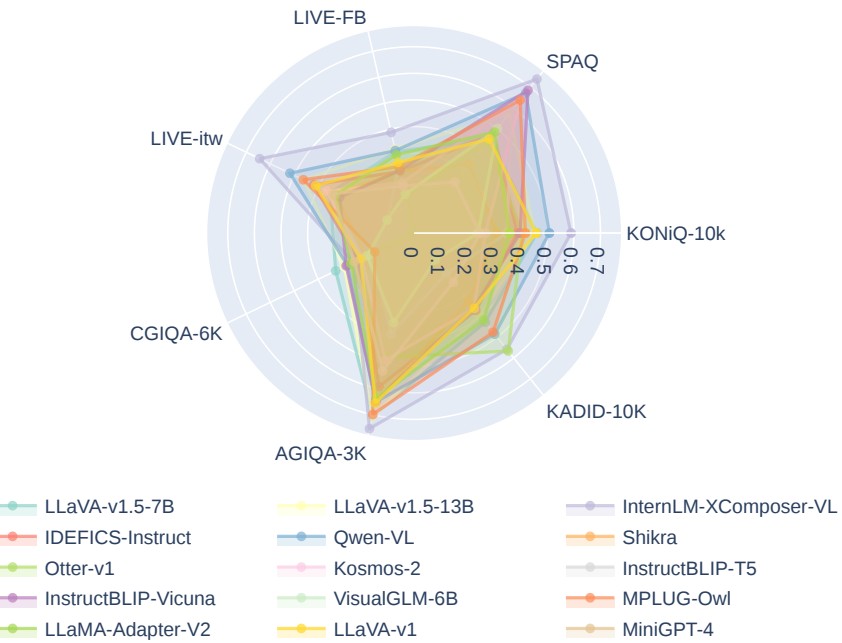

Figure 14: Radar chart for the **Assessment** ability, where the performance of all MLLMs is presented by an average of *SRCC/PLCC* metrics. See Tab. 4 for the respective numerical results.

Table 10: Comparison between *high↔low* and *good↔poor* on InstructBLIP (*Flan-T5-XL*).

| Dataset Type | | In-the-wild | | | | Generated | Artificial |
|---|---|---|---|---|---|---|---|
| Model / Dataset | tokens | *KONiQ-10k* | *SPAQ* | *LIVE-FB* | *LIVE-itw* | *AGIQA-3K* | *KADID-10K* |
| InstructBLIP *(Flan-T5-XL)* | *good↔poor* | 0.288/0.289 | 0.581/**0.618** | 0.221/0.231 | 0.017/0.020 | 0.264/0.281 | **0.264/0.220** |
| InstructBLIP *(Flan-T5-XL)* | *high↔low* | **0.334/0.362** | **0.582**/0.599 | **0.248/0.267** | **0.113/0.113** | **0.378/0.400** | 0.211/0.179 |

such results are obtained via no direct alignment with human perception and from the simplest prompts (*"Rate the quality of the image."*), they suggest the exciting underlying abilities of general intelligence to naturally understand "quality" via their vast training data. However, the performance are still yet to be accurate on finer-grained situations, such as LIVE-FB, which is constructed by more than 95% high-quality images (*i.e. quality score > 50/100*), or CGIQA-6K, made up entirely by relatively high-quality images collected from video games or movies. This suggests that MLLMs still need to improve the measurability on their predictions through well-designed fine-tuning.

**A Deep Dive Into the Probabilities:**

**(A) Softmax *vs* Argmax:**

In the first part of the deep dive, we quantitatively evaluate the correlation with human perception on a simple `argmax` strategy between *good↔bad* and our proposed softmax strategy. In Tab. 9, we confirm that for all MLLMs on all IQA datasets, the more measurable `softmax` strategy predicts better than the `argmax` strategy, which degenerates into only two scores, 0 and 1. Though the result is generally expected, the experiments validate that MLLMs have quantitative **assessment** ability hidden behind their word outputs, and prove the effectiveness of our softmax-based IQA strategy.

**(B) [For T5-based InstructBLIP] *high↔low vs good↔poor*:**

We further conduct a special study for InstructBLIP (*Flan-T5-XL*). With a different LLM as language backbone, even pre-trained with the same settings, the T5-version of InstructBLIP tends to predict more *high↔low* than *good↔poor*, different from its *Vicuna-7B*-based counterpart. The experimental results in Tab 10 validate that the more probable *high↔low* tokens are more competitive in IQA than *good↔bad* tokens, suggesting that top-frequency tokens are more quality-distinctive.

Table 11: Evaluation results on the *synonym ensemble* strategy for the (A3) **Assessment** ability on MLLMs with top-5 results in the default A3 leaderboard of the Q-Bench. After *ensemble*, the rankings among them are not changed. Metrics are *SRCC/PLCC*.

| Dataset Type | In-the-wild | | | | Generated | | Artificial | *Average* |
|---|---|---|---|---|---|---|---|---|
| **Prompt / Dataset** | *KONiQ-10k* | *SPAQ* | *LIVE-FB* | *LIVE-itw* | *CGIQA-6K* | *AGIQA-3K* | *KADID-10K* | |
| **LLaVA-v1.5 (*Vicuna-v1.5-13B*)** | | | | | | | | |
| *good↔poor* | 0.448/0.460 | 0.563/0.584 | 0.310/0.339 | 0.445/0.481 | 0.285/0.297 | 0.664/0.754 | 0.390/0.400 | 0.444/0.473 |
| *fine↔bad* | 0.449/0.487 | 0.583/0.597 | 0.316/0.360 | 0.466/0.513 | 0.349/0.365 | 0.650/0.749 | 0.425/0.437 | 0.463/**0.501** |
| *high↔low* | 0.456/0.482 | 0.529/0.553 | 0.286/0.306 | 0.489/0.513 | 0.276/0.284 | 0.683/0.752 | 0.316/0.331 | 0.434/0.460 |
| *good+high↔poor+low* | 0.462/0.484 | 0.548/0.573 | 0.303/0.327 | 0.480/0.509 | 0.283/0.294 | 0.687/0.763 | 0.350/0.363 | 0.445/0.473 |
| *good+fine↔poor+bad* | 0.463/0.483 | 0.579/0.596 | 0.321/0.356 | 0.467/0.505 | 0.326/0.339 | 0.670/0.762 | 0.420/0.426 | **0.464**/0.495 |
| *good+high+fine↔poor+low+bad* | 0.474/0.498 | 0.565/0.588 | 0.314/0.345 | 0.488/0.521 | 0.311/0.322 | 0.692/0.771 | 0.382/0.392 | 0.461/0.491 |
| **LLaVA-v1.5 (*Vicuna-v1.5-7B*)** | | | | | | | | |
| *good↔poor* | 0.463/0.459 | 0.443/0.467 | 0.305/0.321 | 0.344/0.358 | 0.321/0.333 | 0.672/0.738 | 0.417/0.440 | 0.424/0.445 |
| *fine↔bad* | 0.453/0.469 | 0.457/0.482 | 0.258/0.288 | 0.303/0.333 | 0.294/0.302 | 0.558/0.617 | 0.389/0.420 | 0.388/0.416 |
| *high↔low* | 0.474/0.476 | 0.370/0.386 | 0.261/0.262 | 0.432/0.429 | 0.266/0.269 | 0.669/0.716 | 0.304/0.331 | 0.397/0.410 |
| *good+high↔poor+low* | 0.491/0.491 | 0.416/0.436 | 0.293/0.300 | 0.696/0.751 | 0.413/0.416 | 0.298/0.304 | 0.359/0.389 | 0.424/0.441 |
| *good+fine↔poor+bad* | 0.482/0.482 | 0.461/0.485 | 0.300/0.320 | 0.644/0.708 | 0.339/0.357 | 0.327/0.336 | 0.425/0.451 | 0.425/0.449 |
| *good+high+fine↔poor+low+bad* | 0.512/0.513 | 0.443/0.465 | 0.303/0.315 | 0.408/0.415 | 0.318/0.324 | 0.697/0.752 | 0.392/0.421 | **0.439/0.458** |
| **mPLUG-Owl (*LLaMA-7B*)** | | | | | | | | |
| *good↔poor* | 0.409/0.427 | 0.634/0.644 | 0.241/0.271 | 0.437/0.487 | 0.148/0.180 | 0.687/0.711 | 0.466/0.486 | 0.432/0.458 |
| *fine↔bad* | 0.357/0.398 | 0.622/0.636 | 0.260/0.290 | 0.422/0.475 | 0.178/0.224 | 0.606/0.646 | 0.536/0.534 | 0.426/0.458 |
| *high↔low* | 0.353/0.369 | 0.610/0.624 | 0.176/0.187 | 0.436/0.464 | 0.110/0.124 | 0.662/0.663 | 0.361/0.378 | 0.387/0.401 |
| *good+high↔poor+low* | 0.382/0.402 | 0.626/0.642 | 0.208/0.228 | 0.446/0.483 | 0.125/0.144 | 0.684/0.697 | 0.409/0.432 | 0.411/0.432 |
| *good+fine↔poor+bad* | 0.403/0.430 | 0.635/0.645 | 0.260/0.292 | 0.444/0.493 | 0.172/0.213 | 0.664/0.694 | 0.525/0.527 | **0.443/0.471** |
| *good+high+fine↔poor+low+bad* | 0.395/0.421 | 0.633/0.647 | 0.233/0.258 | 0.455/0.496 | 0.147/0.173 | 0.685/0.704 | 0.463/0.483 | 0.430/0.455 |
| **Qwen-VL (*QwenLM*)** | | | | | | | | |
| *good↔poor* | 0.470/0.546 | 0.676/0.669 | 0.298/0.339 | 0.504/0.532 | 0.273/0.284 | 0.617/0.686 | 0.486/0.486 | 0.475/0.506 |
| *fine↔bad* | 0.467/0.507 | 0.352/0.365 | 0.205/0.238 | 0.451/0.472 | 0.188/0.185 | 0.599/0.627 | 0.354/0.378 | 0.374/0.396 |
| *high↔low* | 0.531/0.578 | 0.626/0.616 | 0.281/0.290 | 0.574/0.560 | 0.286/0.314 | 0.637/0.692 | 0.332/0.344 | 0.467/0.485 |
| *good+high↔poor+low* | 0.539/0.600 | 0.684/0.673 | 0.299/0.324 | 0.565/0.568 | 0.306/0.330 | 0.660/0.721 | 0.414/0.422 | **0.495/0.520** |
| *good+fine↔poor+bad* | 0.495/0.558 | 0.596/0.581 | 0.264/0.307 | 0.521/0.548 | 0.270/0.270 | 0.640/0.691 | 0.435/0.449 | 0.460/0.486 |
| *good+high+fine↔poor+low+bad* | 0.541/0.600 | 0.632/0.617 | 0.286/0.316 | 0.570/0.577 | 0.301/0.318 | 0.664/0.719 | 0.416/0.429 | 0.487/0.511 |
| **InternLM-XComposer-VL (*InternLM*)** | | | | | | | | |
| *good↔poor* | 0.564/0.615 | 0.730/0.750 | 0.360/0.416 | 0.612/0.676 | 0.243/0.265 | 0.732/0.775 | 0.546/0.572 | 0.541/0.581 |
| *fine↔bad* | 0.546/0.597 | 0.720/0.736 | 0.341/0.389 | 0.626/0.671 | 0.213/0.227 | 0.681/0.708 | 0.494/0.479 | 0.517/0.544 |
| *high↔low* | 0.543/0.590 | 0.704/0.720 | 0.331/0.372 | 0.612/0.656 | 0.223/0.251 | 0.716/0.755 | 0.490/0.500 | 0.517/0.549 |
| *good+high↔poor+low* | 0.564/0.613 | 0.723/0.743 | 0.354/0.405 | 0.621/0.676 | 0.238/0.264 | 0.734/0.775 | 0.522/0.546 | 0.537/0.575 |
| *good+fine↔poor+bad* | 0.573/0.626 | 0.735/0.755 | 0.366/0.420 | 0.629/0.687 | 0.236/0.260 | 0.732/0.771 | 0.531/0.551 | **0.543/0.581** |
| *good+high+fine↔poor+low+bad* | 0.571/0.621 | 0.728/0.748 | 0.360/0.410 | 0.629/0.683 | 0.236/0.261 | 0.734/0.773 | 0.521/0.538 | 0.540/0.576 |

### A.3.5 *Synonym Ensemble*: FURTHER IMPROVING IQA ABILITY (A3) FOR MLLMs

As shown in Table 11, the *synonym ensemble* strategy (as proposed in Eq. 4) on top-5 methods (*i.e.* InternLM-XComposer-VL, QWen-VL, LLaVA-v1.5 (*13B*), mPLUG-Owl, and LLaVA-v1.5 (*7B*)) can in average lead to up to 2% accuracy improvement (*in average 1.3%*). We believe it is a useful boost to improve the performance of MLLMs on IQA task.

Nevertheless, we also notice that different MLLMs perform best with different specific prompt combos. For example, the *good+fine↔poor+bad* performs best on InternLM-XComposer-VL, but comes with reduced accuracy on QWen-VL compared with only *good↔poor*. While *good↔poor* is proved *overall best single word pair* for the evaluation and shows stable results across MLLMs, we decide to keep the current strategy in Q-Bench to evaluate MLLMs.

## B  STATEMENT ON DATA CONTAMINATION

The Q-bench contains three tasks, where the first two tasks, (A1) **perception** and (A2) **description**, are evaluated with our own datasets proposed with the paper. For these two tasks, the questions, answers, or low-level descriptions in the two datasets are not seen by any existing MLLMs. Half of LLVisionQA (*i.e.* the `test` subset) and full of LLDescribe labels are kept private, to avoid being added to the training sets of any MLLMs. We hope that this measure will allow Q-Bench to have long-term significance as an indicator of low-level visual abilities.

For the third task, (A3) **assessment**, the situation is a bit more complicated. For open-source models as tested, almost all of them have provided their technical reports, and as far as we know, **no** image quality assessment (IQA) dataset has participated in the **multi-modality training stages** of them.

While text knowledge about image quality assessment should have been injected to them (*e.g.* a blurry image is a low quality image) during their **pure-language training stages**, we think this should not be regarded as data contamination for IQA, because the images cannot be seen by a language model. Instead, they are important knowledge for MLLMs to better link particular visual attributes (*blur*) to human opinions (*quality*), which motivates us to explore MLLMs for these tasks.

## C LIMITATIONS AND DISCUSSIONS

In Section A.3.2, we observed that MLLMs frequently respond with '*yes*' to *Yes-or-No* questions. It's worth noting that the current **LLVisionQA** dataset is skewed, with 62% of its questions being Yes-questions and only 38% being No-questions. This imbalance could introduce biases when comparing various MLLMs. To fully address this, we aim to balance the dataset by preparing a reversed version for each question in our subsequent work, ensuring a less biased evaluation.

For the **description** task, we acknowledge that judging whether a description matches the gold description is a subjective process, which may not have an absolute standard. Even when evaluated by humans, the scores rated for the MLLM descriptions are subject to individual differences. Though we have employed the 5-round GPT-assisted evaluation protocol, which could be the most reliable and reproducible way at present, it may still unavoidably contain hallucinations (from GPT). We will continue to explore how to design a more reliable evaluation protocol for the low-level visual description task in our follow-up works.

While the proposed **Q-Bench** has offered a comprehensive evaluation on the low-level visual capabilities of MLLMs, it does not provide direct guidance on enhancing these capabilities. As our next steps, we intend to progressively scale up the **LLDescribe** and **LLVisionQA** datasets to eventually allow a reliable *low-level visual instruction tuning* process that can further improve the low-level abilities for MLLMs.

