# OpenReview forum: "Q-Bench: A Benchmark for General-Purpose Foundation Models on Low-level Vision"
_ICLR.cc/2024/Conference — ICLR 2024 spotlight_

### Official Review · Reviewer_rhgp · 2023-10-31

**Soundness:** 3 good
**Presentation:** 3 good
**Contribution:** 3 good
**Rating:** 8
**Confidence:** 4

**Summary:**

This is a benchmark paper on the performance of low-level visual perception and understanding of MLLMs. To this end, the paper collected and annotated a new benchmark dataset Q-Bench including 1) LLVisionQA about the low-level perception of 2990 images; 2) LLDescribe about the description of image quality of 499 images; 3) how to align visual quality scores with people perception. The paper evaluated 10 recent public available MLLM on Q-Bench. The evaluation indicates current MLLMs have decent low-level abilities yet still a long way to go for general low-level visual assessment.

**Strengths:**

The benchmark of current MLLMs’ abilities to access low-level image quality appears a quite interesting topic to me. The paper presented a thorough and in-depth evaluation and the finding may inspire some further research. The evaluation design of Q-Bench makes sense. The paper is well-written.

**Weaknesses:**

This is a descent benchmark paper on an interesting topic. I would recommend acceptance given the performance of GPT-4V on Q-Bench is provided in the revised version.

**Questions:**

It is definitely a must-to-ask question that how the performance of GPT-4-Vision on Q-Bench, which shall establish the baseline for commercial SotA MLLM.

---

> ### Author Response · Authors · 2023-11-11
> **Thank you for your recognition and advice! Here is GPT-4V (and human) performance on Q-Bench.**
>
> We sincerely thank you for the reviewer's recognition on our work, and highly agree with the concerns and suggestions. The additional results as required are provided as follows.
>
> ## Question:
> It is definitely a must-to-ask question that how the performance of GPT-4-Vision on Q-Bench, which shall establish the baseline for commercial SotA MLLM.
>
> ## Answer:
>
> Thank you for your kind suggestion. We totally agree with the opinion of the reviewer that, the performance of GPT-4Vision (GPT-4V) is a must-to-ask question in our benchmark. Therefore, since the day it releases public use, we have been discovering the feasibility and testing the GPT-4V on three sub-tasks of Q-Bench. They are analyzed as follows:
>
> ### 1. The perception task [Feasible]
>
> The task is based on Multi-Choice Questions (MCQ), with objectively-defined correct choices for each question. We primarily test with GPT-4V and make sure they are very good answering MCQs, and will directly give a choice A/B/C/D given the input. Therefore, we test it as follows:
>
> **Example**:
>
> [Image Upload] *What level of blurriness does the background skyscrapers of this image have?*
>
> *Choose between one of the options as follows:*
>
> *A. Slight*
>
> *B. Severe*
>
> *C. Moderate*
>
> And GPT-4V returns *C. Moderate* for this question.
>
> With this method, we evaluate the quantitative result of GPT-4V on the low-level perception task as follows (*This is tested on the `test` subset of LLVisionQA, with 1495 image-question-answers. The answers in this subset will **never be released to public**, so as to avoid contamination and keep the fairest comparison among methods.*)
>
> |**Participant Name** | yes-or-no | what | how | distortion | others | in-context distortion | in-context others | overall |
> | - | - | - | - | - | - | -| - | -|
> | InternLM-XComposer-VL (best open-source) | 0.6843  | 0.6204  | 0.6193  | 0.5681  | 0.7041  | 0.5753  | 0.7719  | 0.6435  |
> | GPT-4V | 0.7792 | 0.7918 | 0.6268 | 0.7058 | 0.7303 | 0.7466 | 0.7795 | 0.7336 (**+0.0801** to best open-source, *i.e.* InternLM-XComposer-VL)  |
> | human-1 (without expertise) | 0.8248 | 0.7939 | 0.6029 | 0.7562 | 0.7208 | 0.7637 | 0.7300 | 0.7431 (+0.0095 to GPT-4V)  |
> | human-2 (with expertise) | **0.8431** | **0.8894** | **0.7202** | **0.7965** | **0.7947** | **0.8390** | **0.8707** | **0.8174** (+0.0838 to GPT-4V)  |
>
> As is shown in the table, GPT-4V can already reach **the level of a non-expert human** (who had not participated in related studies or jobs before), outperforming all open-source approaches by a large margin, but still notably behind a experienced human (who had participated in other related studies, but independent to our author team.)
>
> Nevertheless, as the data used in GPT-4V is a **black box**, related data (low-level visual recognition, image quality assessment) may have been collected and used for training this powerful model. Thus, the comparison is kind of "*not equal*" to the open-source methods, which are confirmed without direct instruction tuning on low-level-related data. Instead, this exciting result points a hopeful future for open-source MLLMs to also reach the same ability, with specifically-designed tuning for low-level visual perception.
>
> ### 2. The description task.  [Not Yet Feasible]
>
> As this task needs GPT itself to evaluate and compare accuracy between two descriptions, it might not be feasible to test GPT-4V on this task in this stage, as some biases might happen (especially in comparison with other MLLMs). We will seek other feasible evaluation protocols that can test GPT-4V more reliably.
>
> ### 3. The assessment task. [Not Yet Feasible]
>
> As is mentioned in our methodology, due to the requirement of a quantifiable output, the assessment task is based on next-word probabilities in an **auto-completion**-like task. Regretfully, at present, providing `log probabilities` of words are disabled for all GPT variants (including the initial versions), including both the web interface and the API. We look forward to seeing this feature return to GPT API one day and we will test its assessment performance then.
>
> In summary, we **have tested the low-level visual perception ability of GPT-4V**, and validate its advantage towards open-source models (though still improvable to replace professional human).  We will continue to figure out new strategies to evaluate GPT-4V in more low-level visual tasks in the future.
>
> Thank you again for providing this advice to test GPT-4V on Q-Bench.
>
> Best Regards,
>
> Authors of the Bench

---

### Official Review · Reviewer_1bay · 2023-11-01

**Soundness:** 3 good
**Presentation:** 3 good
**Contribution:** 3 good
**Rating:** 8
**Confidence:** 5

**Summary:**

The rapidly developed Multi-modality Large Language Models (MLLMs) have progressed greatly. However, there is still an inadequacy in assessing the abilities of MLLMs on low-level visual perception and understanding. Accordingly, this paper proposed Q-Bench, a holistic benchmark of MLLMs for low-level visual perception, low-level visual description, and overall visual quality assessment. Q-Bench constructs solid benchmark datasets and reasonable evaluation pipelines, including LLVisionQA for low-level perception ability, LLDescribe for low-level description ability, and a compounded IQA dataset for image quality assessment with an innovative softmax-based strategy. Q-Bench provides a viable solution to reveal the potential of MLLMs for low-level visual understanding.

**Strengths:**

1. Q-Bench is composed of benchmark datasets and reasonable evaluation pipelines for low-level perception, low-level description, and image quality assessment. The contributions are solid.
2. The experiments and study are extensive and convincing.
3. The Q-Bench is well presented, and the details are clear.

**Weaknesses:**

1. In the evaluation of LLDescribe, the softmax is calculated between good and poor. How about also considering their synonyms, e.g., great, excellent, fine, bad, low, etc? Here is a possible solution: (1) merge the logits of great, excellent, fine into good, and bad, low into poor. (2) calculate the final score with merged logits.
2. As a benchmark paper, the author may update the recent SOTA MLLMs into the leadboards of this paper, e.g., QWen-VL [1], InternLM-XComposer [2], and LLaVA-1.5 [3].

[1] Qwen-VL: A Versatile Vision-Language Model for Understanding, Localization, Text Reading, and Beyond
[2] InternLM-XComposer: A Vision-Language Large Model for Advanced Text-image Comprehension and Composition
[3] Improved Baselines with Visual Instruction Tuning

**Questions:**

See weakness.

---

> ### Author Response · Authors · 2023-11-11
> **(Thread 4/4) Thank you for your kind support and suggestions!**
>
> [Thread 4/4]
>
> ## Question 2 (Q2)
>
> As a benchmark paper, the author may update the recent SOTA MLLMs into the leadboards of this paper, e.g., QWen-VL [1], InternLM-XComposer [2], and LLaVA-1.5 [3].
>
> ## Answer for Q2 (Part 3)
>
>
> ### Updated Results for (A3): Assessment
>
> | **Model Name**|KoNIQ-10k | SPAQ| LIVE-FB| LIVE-itw| CGIQA-6K| AGIQA-3K| KADID-10K|Average
> | -| -| -| -| -| -| -| -| -|
> | NIQE | 0.316/0.377 | 0.693/0.669 | 0.211/0.288 | 0.480/0.451 | 0.075/0.056 | 0.562/0.517 | 0.374/0.428 |0.387/0.398|
> | CLIP-ViT-Large-14 | 0.468/0.505 | 0.385/0.389 | 0.218/0.237 | 0.307/0.308 | 0.285/0.290 | 0.436/0.458 | 0.376/0.388 |0.354/0.368|
> | LLaVA-v1.5 (Vicuna-v1.5-7B) | 0.463/0.459 | 0.443/0.467 | 0.305/0.321 | 0.344/0.358 | **0.321/0.333** | 0.672/0.738 | 0.417/0.440 |0.424/0.445|
> | LLaVA-v1.5 (Vicuna-v1.5-13B) | 0.448/0.460 | 0.563/0.584 | 0.310/0.339 | 0.445/0.481 | 0.285/0.297 | 0.664/0.754 | 0.390/0.400 |0.444/0.474|
> | InternLM-XComposer-VL (InternLM) | **0.564/0.615** | **0.730/0.750** | **0.360/0.416** | **0.612/0.676** | 0.243/0.265 | **0.732/0.775** | 0.546/0.572 |**0.541/0.581**|
> | IDEFICS-Instruct (LLaMA-7B) | 0.375/0.400 | 0.474/0.484 | 0.235/0.24 | 0.409/0.428 | 0.244/0.227 | 0.562/0.622 | 0.370/0.373 |0.381/0.396|
> | Qwen-VL (QwenLM) | 0.470/0.546 | 0.676/0.669 | 0.298/0.338 | 0.504/0.532 | 0.273/0.284 | 0.617/0.686 | 0.486/0.486 |0.475/0.506|
> | Shikra (Vicuna-7B) | 0.314/0.307 | 0.32/0.337 | 0.237/0.241 | 0.322/0.336 | 0.198/0.201 | 0.640/0.661 | 0.324/0.332 |0.336/0.345|
> | Otter-v1 (MPT-7B) | 0.406/0.406 | 0.436/0.441 | 0.143/0.142 | -0.008/0.018 | 0.254/0.264 | 0.475/0.481 | **0.557/0.577** |0.323/0.333|
> | Kosmos-2 | 0.255/0.281 | 0.644/0.641 | 0.196/0.195 | 0.358/0.368 | 0.210/0.225 | 0.489/0.491 | 0.359/0.365 |0.359/0.367|
> | InstructBLIP (Flan-T5-XL) | 0.334/0.362 | 0.582/0.599 | 0.248/0.267 | 0.113/0.113 | 0.167/0.188 | 0.378/0.400 | 0.211/0.179 |0.290/0.301|
> | InstructBLIP (Vicuna-7B) | 0.359/0.437 | 0.683/0.689 | 0.200/0.283 | 0.253/0.367 | 0.263/0.304 | 0.629/0.663 | 0.337/0.382 |0.389/0.446|
> | VisualGLM-6B (GLM-6B) | 0.247/0.234 | 0.498/0.507 | 0.146/0.154 | 0.110/0.116 | 0.209/0.183 | 0.342/0.349 | 0.127/0.131 |0.240/0.239|
> | mPLUG-Owl (LLaMA-7B) | 0.409/0.427 | 0.634/0.644 | 0.241/0.271 | 0.437/0.487 | 0.148/0.180 | 0.687/0.711 | 0.466/0.486 |0.432/0.458|
> | LLaMA-Adapter-V2 | 0.354/0.363 | 0.464/0.506 | 0.275/0.329 | 0.298/0.360 | 0.257/0.271 | 0.604/0.666 | 0.412/0.425 |0.381/0.417|
> | LLaVA-v1 (Vicuna-13B) | 0.462/0.457 | 0.442/0.462 | 0.264/0.280 | 0.404/0.417 | 0.208/0.237 | 0.626/0.684 | 0.349/0.372 |0.394/0.416|
> | MiniGPT-4 (Vicuna-13B) | 0.239/0.257 | 0.238/0.253 | 0.170/0.183 | 0.339/0.340 | 0.252/0.246 | 0.572/0.591 | 0.239/0.233 |0.293/0.300|

---

> ### Author Response · Authors · 2023-11-11
> **(Thread 3/4) Thank you for your kind support and suggestions!**
>
> [Thread 3/4]
> ## Question 2 (Q2)
>
> As a benchmark paper, the author may update the recent SOTA MLLMs into the leadboards of this paper, e.g., QWen-VL [1], InternLM-XComposer [2], and LLaVA-1.5 [3].
>
> ## Answer for Q2 (Part 2)
>
> ### Updated Results for (A2): Description
>
> Abbreviations for dimensions: *comp: completeness, prec: precision, rele: relevance*
>
> | **Model Name** | p_{0, comp} | p_{0, comp} | p_{2, comp} | s_{compl} | p_{0, prec} | p_{0, prec} | p_{2, prec} | s_{prec} | p_{0, rele} | p_{0, rele} | p_{2, rele} | s_{rele} | s_{sum} |
> | - | - | - | - | - | - | - | - | - | - | - | - | - | - |
> | LLaVA-v1.5 (Vicuna-v1.5-13B) | 27.68%  | 53.78%  | 18.55%  | 0.91/2.00 | 25.45%  | 21.47%  | 53.08%  | 1.28/2.00 | 6.31%   | 58.75%   | 34.94%   | 1.29/2.00 | 3.47/6.00 |
> | InternLM-XComposer-VL (InternLM) | 19.94%  | 51.82%  | 28.24%  | 1.08/2.00 | 22.59%  | 28.99%  | 48.42%  | 1.26/2.00 | 1.05%   | 10.62%   | 88.32%   | 1.87/2.00 | 4.21/6.00 |
> | IDEFICS-Instruct (LLaMA-7B) | 28.91%  | 59.16%  | 11.93%  | 0.83/2.00 | 34.68%  | 27.86%  | 37.46%  | 1.03/2.00 | 3.90%   | 59.66%   | 36.44%   | 1.33/2.00 | 3.18/6.00 |
> | Qwen-VL (QwenLM)           | 26.34%  | 49.13%  | 24.53%  | 0.98/2.00 | 50.62%  | 23.44%  | 25.94%  | 0.75/2.00 | 0.73%   | 35.56%   | 63.72%   | 1.63/2.00 | 3.36/6.00 |
> | Shikra (Vicuna-7B)         | 21.14%  | 68.33%  | 10.52%  | 0.89/2.00 | 30.33%  | 28.30%  | 41.37%  | 1.11/2.00 | 1.14%   | 64.36%   | 34.50%   | 1.33/2.00 | 3.34/6.00 |
> | Otter-v1 (MPT-7B)          | 22.38%  | 59.36%  | 18.25%  | 0.96/2.00 | 40.68%  | 35.99%  | 23.33%  | 0.83/2.00 | 1.95%   | 13.2%    | 84.85%   | 1.83/2.00 | 3.61/6.00 |
> | Kosmos-2                   | 8.76%   | 70.91%  | 20.33%  | 1.12/2.00 | 29.45%  | 34.75%  | 35.81%  | 1.06/2.00 | 0.16%   | 14.77%   | 85.06%   | 1.85/2.00 | 4.03/6.00 |
> | InstructBLIP (Flan-T5-XL)  | 23.16%  | 66.44%  | 10.40%  | 0.87/2.00 | 34.85%  | 26.03%  | 39.12%  | 1.04/2.00 | 14.71%  | 59.87%   | 25.42%   | 1.11/2.00 | 3.02/6.00 |
> | InstructBLIP (Vicuna-7B)   | 29.73%  | 61.47%  | 8.80%   | 0.79/2.00 | 27.84%  | 23.52%  | 48.65%  | 1.21/2.00 | 27.40%  | 61.29%   | 11.31%   | 0.84/2.00 | 2.84/6.00 |
> | VisualGLM-6B (GLM-6B)      | 30.75%  | 56.64%  | 12.61%  | 0.82/2.00 | 38.64%  | 26.18%  | 35.18%  | 0.97/2.00 | 6.14%   | 67.15%   | 26.71%   | 1.21/2.00 | 2.99/6.00 |
> | mPLUG-Owl (LLaMA-7B)       | 28.28%  | 37.69%  | 34.03%  | 1.06/2.00 | 26.75%  | 18.18%  | 55.07%  | 1.28/2.00 | 3.03%   | 33.82%   | 63.15%   | 1.6/2.00  | 3.94/6.00 |
> | LLaMA-Adapter-V2           | 30.44%  | 53.99%  | 15.57%  | 0.85/2.00 | 29.41%  | 25.79%  | 44.8%   | 1.15/2.00 | 1.50%   | 52.75%   | 45.75%   | 1.44/2.00 | 3.45/6.00 |
> | LLaVA-v1 (Vicuna-13B)      | 34.10%  | 40.52%  | 25.39%  | 0.91/2.00 | 30.02%  | 15.15%  | 54.83%  | 1.25/2.00 | 1.06%   | 38.03%   | 60.91%   | 1.6/2.00  | 3.76/6.00 |
> | MiniGPT-4 (Vicuna-13B)     | 34.01%  | 32.15%  | 33.85%  | 1.00/2.00 | 29.20%  | 15.27%  | 55.53%  | 1.26/2.00 | 6.88%   | 45.65%   | 47.48%   | 1.41/2.00 | 3.67/6.00 |

---

> ### Author Response · Authors · 2023-11-11
> **(Thread 2/4) Thank you for your kind support and suggestions!**
>
> [Thread 2/4]
> ## Question 2 (Q2)
>
> As a benchmark paper, the author may update the recent SOTA MLLMs into the leadboards of this paper, e.g., QWen-VL [1], InternLM-XComposer [2], and LLaVA-1.5 [3].
>
> ## Answer for Q2 (Part 1)
>
> ### Updated Results for (A1): Perception
>
> In the updated version, we split the LLVisionQA set into `dev` set (half) and `test` set (half). The former will be released for future MLLMs to test locally and develop their methods, and the latter will not be released to avoid contamination (future methods can evaluate on `test` via our service).
>
> #### 1. Accuracies on `dev`
>
> |**Model Name** | yes-or-no | what | how | distortion | others | in-context distortion | in-context others | overall |
> | - | - | - | - | - | - | -| - | -|
> | random guess              | 0.5000  | 0.2786  | 0.3331  | 0.3789  | 0.3848  | 0.3828  | 0.3582  | 0.3780  |
> | LLaVA-v1.5 (Vicuna-v1.5-7B) | 0.6636  | 0.5819  | 0.5051  | 0.4942  | 0.6574  | 0.5461  | 0.7061  | 0.5866  |
> | LLaVA-v1.5 (Vicuna-v1.5-13B) | 0.6527  | 0.6438  | 0.5659  | 0.5603  | 0.6713  | 0.6118  | 0.6735  | 0.6214  |
> | InternLM-XComposer-VL (InternLM) | 0.6945  | 0.6527  | 0.6085  | 0.6167  | 0.7014  | 0.5691  | 0.7510  | **0.6535**  |
> | IDEFICS-Instruct (LLaMA-7B) | 0.5618  | 0.4469  | 0.4402  | 0.4280  | 0.5417  | 0.4474  | 0.5633  | 0.4870  |
> | Qwen-VL (QwenLM)           | 0.6309  | 0.5819  | 0.5639  | 0.5058  | 0.6273  | 0.5789  | 0.7388  | 0.5940  |
> | Shikra (Vicuna-7B)         | 0.6564  | 0.4735  | 0.4909  | 0.4883  | 0.5949  | 0.5000  | 0.6408  | 0.5465  |
> | Otter-v1 (MPT-7B)          | 0.5709  | 0.4071  | 0.3955  | 0.4222  | 0.4931  | 0.4408  | 0.5265  | 0.4635  |
> | InstructBLIP (Flan-T5-XL)  | 0.6764  | 0.5996  | 0.5598  | 0.5623  | 0.6551  | 0.5822  | 0.6939  | 0.6147  |
> | InstructBLIP (Vicuna-7B)   | 0.7164  | 0.5265  | 0.4381  | 0.4864  | 0.6250  | 0.5559  | 0.6490  | 0.5672  |
> | VisualGLM-6B (GLM-6B)      | 0.6018  | 0.5420  | 0.4625  | 0.5175  | 0.5440  | 0.5362  | 0.5714  | 0.5378  |
> | mPLUG-Owl (LLaMA-7B)       | 0.6600  | 0.5487  | 0.4402  | 0.5136  | 0.5509  | 0.5428  | 0.6571  | 0.5538  |
> | LLaMA-Adapter-V2           | 0.6618  | 0.5929  | 0.5213  | 0.5739  | 0.5625  | 0.6316  | 0.6490  | 0.5946  |
> | LLaVA-v1 (Vicuna-13B)      | 0.5400  | 0.5310  | 0.5538  | 0.4864  | 0.5463  | 0.5559  | 0.6327  | 0.5418  |
> | MiniGPT-4 (Vicuna-13B)     | 0.5582  | 0.5022  | 0.4037  | 0.4202  | 0.4838  | 0.5197  | 0.6122  | 0.4903  |
>
>
>
> #### 2. Accuracies on `test`
>
> - *GPT-4V and human*:
>
> |**Name** | yes-or-no | what | how | distortion | others | in-context distortion | in-context others | overall |
> | - | - | - | - | - | - | -| - | -|
> | GPT-4V (Close-Source Model)   | 0.7792  | 0.7918  | 0.6268  | 0.7058  | 0.7303  | 0.7466  | 0.7795  | 0.7336  |
> | human-1 (without expertise)         | 0.8248  | 0.7939  | 0.6029  | 0.7562  | 0.7208  | 0.7637  | 0.7300  | 0.7431  |
> | human-2 (with expertise)            | 0.8431  | 0.8894  | 0.7202  | 0.7965  | 0.7947  | 0.8390  | 0.8707  | 0.8174  |
>
> GPT-4V is primarily a human without expertise on low-level visual perception. Please see responses to reviewer `rhgp` for more analysis.
>
>
>
>
> - *Open-source MLLMs*:
>
> |**Model Name** | yes-or-no | what | how | distortion | others | in-context distortion | in-context others | overall |
> | - | - | - | - | - | - | -| - | -|
> | random guess              | 0.5000  | 0.2848  | 0.3330  | 0.3724  | 0.3850  | 0.3913  | 0.3710  | 0.3794  |
> | LLaVA-v1.5 (Vicuna-v1.5-7B) | 0.6460  | 0.5922  | 0.5576  | 0.4798  | 0.6730  | 0.5890  | 0.7376  | 0.6007  |
> | LLaVA-v1.5 (Vicuna-v1.5-13B) | 0.6496  | 0.6486  | 0.5412  | 0.5355  | 0.6659  | 0.5890  | 0.7148  | 0.6140  |
> | InternLM-XComposer-VL (InternLM) | 0.6843  | 0.6204  | 0.6193  | 0.5681  | 0.7041  | 0.5753  | 0.7719  | **0.6435**  |
> | IDEFICS-Instruct (LLaMA-7B) | 0.6004  | 0.4642  | 0.4671  | 0.4038  | 0.5990  | 0.4726  | 0.6477  | 0.5151  |
> | Qwen-VL (QwenLM)           | 0.6533  | 0.6074  | 0.5844  | 0.5413  | 0.6635  | 0.5822  | 0.7300  | 0.6167  |
> | Shikra(Vicuna-7B)          | 0.6909  | 0.4793  | 0.4671  | 0.4731  | 0.6086  | 0.5308  | 0.6477  | 0.5532  |
> | Otter-v1 (MPT-7B)          | 0.5766  | 0.3970  | 0.4259  | 0.4212  | 0.4893  | 0.4760  | 0.5417  | 0.4722  |
> | InstructBLIP (Flan-T5-XL)  | 0.6953  | 0.5900  | 0.5617  | 0.5731  | 0.6551  | 0.5651  | 0.7121  | 0.6194  |
> | InstructBLIP (Vicuna-7B)   | 0.7099  | 0.5141  | 0.4300  | 0.4500  | 0.6301  | 0.5719  | 0.6439  | 0.5585  |
> | VisualGLM-6B (GLM-6B)      | 0.6131  | 0.5358  | 0.4403  | 0.4856  | 0.5489  | 0.5548  | 0.5779  | 0.5331  |
> | mPLUG-Owl (LLaMA-7B)       | 0.7245  | 0.5488  | 0.4753  | 0.4962  | 0.6301  | 0.6267  | 0.6667  | 0.5893  |
> | LLaMA-Adapter-V2           | 0.6618  | 0.5466  | 0.5165  | 0.5615  | 0.6181  | 0.5925  | 0.5455  | 0.5806  |
> | LLaVA-v1 (Vicuna-13B)      | 0.5712  | 0.5488  | 0.5185  | 0.4558  | 0.5800  | 0.5719  | 0.6477  | 0.5472  |
> | MiniGPT-4 (Vicuna-13B)     | 0.6077  | 0.5033  | 0.4300  | 0.4558  | 0.5251  | 0.5342  | 0.6098  | 0.5177  |

---

> ### Author Response · Authors · 2023-11-11
> **(Thread 1/4) Thank you for your kind support and valuable suggestions!**
>
> [Thread 1/4]
>
> We thank the reviewer for the recognition of our work's contributions, solidity of validations, and presentation.
>
> We also sincerely consider the two suggestions as proposed by the reviewer:
>
> 1. *Include **synonym ensemble** for image quality assessment (IQA).* This can bring general improvements (up to 2.1%) on five top methods on the existing IQA benchmark (task A3 of Q-Bench), while the best synonym combination is not the same for all methods. We will recommend this strategy as an additional boost strategy in our paper. See this thread [Thread 1/4] for details.
>
> 2. *Include **more MLLMs** into Q-Bench.* Holding the same opinion with the reviewer (that a benchmark should be up-to-date), we keep tracking on MLLMs, and have internally **expanded the benchmark to 15 open-source MLLMs**. These internal results are updated here and will be update to our revised paper. See this thread [Thread 2/4] to [Thread 4/4]  for the updated results.
>
>
>
> ## Question 1 (Q1)
>
> In the evaluation of LLDescribe, the softmax is calculated between good and poor. How about also considering their synonyms, e.g., great, excellent, fine, bad, low, etc? Here is a possible solution: (1) merge the logits of great, excellent, fine into good, and bad, low into poor. (2) calculate the final score with merged logits.
>
> ## Answer to Q1
>
> We sincerely thank the reviewer for providing this solution. In short, this can bring improvements.
>
> As shown in the tables below, the results of the synonym ensemble on top-5 methods (InternLM-XComposer-VL, QWen-VL, LLaVA-v1.5-13B, mPLUG-Owl, and LLaVA-v1.5-7B) can in average lead to up to 2% accuracy improvement. We believe it is a very useful suggestion (*esp.* during application) to improve performance of MLLMs on IQA, and we will carefully record it in our paper.
>
> Nevertheless, we also notice that different MLLMs perform best with different specific prompt combos, adding difficulties to us when evaluating new models. While *good/poor* (*which is overall the best single word pair*) is only in average 1.3% inferior to ensemble performance and shows the same inter-model ranking, we decide to keep the current strategy in Q-Bench to evaluate MLLMs, but set an independent chapter to **recommend the ensemble strategy while applying MLLMs for IQA.**
>
> Results of different methods are as follows.
>
> ##### 1. LLaVA-v1.5-13B (improves 2.1%)
>
> | **Model Name**| KoNIQ-10k| SPAQ| LIVE-FB| LIVE-itw| AGIQA-3K| CGIQA-6K| KADID-10K| average|
> | -| -| -| -| -| -| -| -| -|
> |pos: ['good'], neg: ['poor'] | 0.448/0.460 | 0.563/0.584 | 0.310/0.339 | 0.445/0.481 | 0.664/0.754 | 0.285/0.297 | 0.390/0.400| 0.444/0.473|
> |pos: ['fine'], neg: ['bad'] | 0.449/0.487 | 0.583/0.597 | 0.316/0.360 | 0.466/0.513 | 0.650/0.749 | 0.349/0.365 | 0.425/0.437| *0.463/0.501 (best single)*|
> |pos: ['high'], neg: ['low'] | 0.456/0.482 | 0.529/0.553 | 0.286/0.306 | 0.489/0.513 | 0.683/0.752 | 0.276/0.284 | 0.316/0.331| 0.434/0.460|
> |pos: ['good', 'high'], neg: ['poor', 'low'] | 0.462/0.484 | 0.548/0.573 | 0.303/0.327 | 0.480/0.509 | 0.687/0.763 | 0.283/0.294 | 0.350/0.363| 0.445/0.473|
> |pos: ['good', 'fine'], neg: ['poor', 'bad'] | 0.463/0.483 | 0.579/0.596 | 0.321/0.356 | 0.467/0.505 | 0.670/0.762 | 0.326/0.339 | 0.420/0.426| **0.464/0.495 (best ensemble)**|
> |pos: ['good', 'high', 'fine'], neg: ['poor', 'low', 'bad'] | 0.474/0.498 | 0.565/0.588 | 0.314/0.345 | 0.488/0.521 | 0.692/0.771 | 0.311/0.322 | 0.382/0.392| 0.461/0.491|
> |pos: ['good', 'high', 'excellent', 'fine'], neg: ['poor', 'low', 'bad'] | 0.452/0.478 | 0.547/0.573 | 0.302/0.328 | 0.486/0.515 | 0.684/0.762 | 0.286/0.297 | 0.361/0.375| 0.445/0.476
>
> ##### 2. LLaVA-v1.5-7B (improves 1.4%)
>
> | **Model Name**| KoNIQ-10k| SPAQ| LIVE-FB| LIVE-itw| AGIQA-3K| CGIQA-6K| KADID-10K| average|
> | -| -| -| -| -| -| -| -| -|
> |pos: ['good'], neg: ['poor'] | 0.463/0.459 | 0.443/0.467 | 0.305/0.321 | 0.344/0.358 | 0.672/0.738 | 0.321/0.333 | 0.417/0.440| *0.424/0.445 (best single)*|
> |pos: ['fine'], neg: ['bad'] | 0.453/0.469 | 0.457/0.482 | 0.258/0.288 | 0.303/0.333 | 0.558/0.617 | 0.294/0.302 | 0.389/0.420| 0.388/0.416|
> |pos: ['high'], neg: ['low'] | 0.474/0.476 | 0.370/0.386 | 0.261/0.262 | 0.432/0.429 | 0.669/0.716 | 0.266/0.269 | 0.304/0.331| 0.397/0.410|
> |pos: ['good', 'high'], neg: ['poor', 'low'] | 0.491/0.491 | 0.416/0.436 | 0.293/0.300 | 0.413/0.416 | 0.696/0.751 | 0.298/0.304 | 0.359/0.389| 0.424/0.441|
> |pos: ['good', 'fine'], neg: ['poor', 'bad'] | 0.482/0.482 | 0.461/0.485 | 0.300/0.320 | 0.339/0.357 | 0.644/0.708 | 0.327/0.336 | 0.425/0.451| 0.425/0.449|
> |pos: ['good', 'high', 'fine'], neg: ['poor', 'low', 'bad'] | 0.512/0.513 | 0.443/0.465 | 0.303/0.315 | 0.408/0.415 | 0.697/0.752 | 0.318/0.324 | 0.392/0.421| **0.439/0.458 (best ensemble)**|
> |pos: ['good', 'high', 'excellent', 'fine'], neg: ['poor', 'low', 'bad'] | 0.480/0.483 | 0.414/0.435 | 0.283/0.291 | 0.410/0.411 | 0.694/0.747 | 0.298/0.301 | 0.373/0.401| 0.422/0.438|
>
> (see follow up)

---

> ### Author Response · Authors · 2023-11-12
> **Follow up for Thread [1/4]: Results of Synonym Ensemble on mPLUG-Owl, Qwen-VL, and InternLM-XComposer-VL**
>
> (following above)
>
> ##### 3. mPLUG-Owl (improves 1.2%)
>
> | **Model Name**| KoNIQ-10k| SPAQ| LIVE-FB| LIVE-itw| AGIQA-3K| CGIQA-6K| KADID-10K| average|
> | -| -| -| -| -| -| -| -| -|
> |pos: ['good'], neg: ['poor'] | 0.409/0.427 | 0.634/0.644 | 0.241/0.271 | 0.437/0.487 | 0.687/0.711 | 0.148/0.180 | 0.466/0.486| *0.432/0.458 (best single)*|
> |pos: ['fine'], neg: ['bad'] | 0.357/0.398 | 0.622/0.636 | 0.260/0.290 | 0.422/0.475 | 0.606/0.646 | 0.178/0.224 | 0.536/0.534| 0.426/0.458|
> |pos: ['high'], neg: ['low'] | 0.353/0.369 | 0.610/0.624 | 0.176/0.187 | 0.436/0.464 | 0.662/0.663 | 0.110/0.124 | 0.361/0.378| 0.387/0.401|
> |pos: ['good', 'high'], neg: ['poor', 'low'] | 0.382/0.402 | 0.626/0.642 | 0.208/0.228 | 0.446/0.483 | 0.684/0.697 | 0.125/0.144 | 0.409/0.432| 0.411/0.432|
> |pos: ['good', 'fine'], neg: ['poor', 'bad'] | 0.403/0.430 | 0.635/0.645 | 0.260/0.292 | 0.444/0.493 | 0.664/0.694 | 0.172/0.213 | 0.525/0.527| **0.443/0.471 (best ensemble)** |
> |pos: ['good', 'high', 'fine'], neg: ['poor', 'low', 'bad'] | 0.395/0.421 | 0.633/0.647 | 0.233/0.258 | 0.455/0.496 | 0.685/0.704 | 0.147/0.173 | 0.463/0.483| 0.430/0.455|
> |pos: ['good', 'high', 'excellent', 'fine'], neg: ['poor', 'low', 'bad'] | 0.374/0.399 | 0.626/0.642 | 0.211/0.230 | 0.447/0.487 | 0.681/0.695 | 0.147/0.170 | 0.441/0.462| 0.418/0.441|
>
> ##### 4. Qwen-VL (improves 1.7%)
>
> | **Model Name**| KoNIQ-10k| SPAQ| LIVE-FB| LIVE-itw| AGIQA-3K| CGIQA-6K| KADID-10K| average|
> | -| -| -| -| -| -| -| -| -|
> |pos: ['good'], neg: ['poor'] | 0.470/0.546 | 0.676/0.669 | 0.298/0.339 | 0.504/0.532 | 0.617/0.686 | 0.273/0.284 | 0.486/0.486| *0.475/0.506 (best single)* |
> |pos: ['fine'], neg: ['bad'] | 0.467/0.507 | 0.352/0.365 | 0.205/0.238 | 0.451/0.472 | 0.599/0.627 | 0.188/0.185 | 0.354/0.378| 0.374/0.396|
> |pos: ['high'], neg: ['low'] | 0.531/0.578 | 0.626/0.616 | 0.281/0.290 | 0.574/0.560 | 0.637/0.692 | 0.286/0.314 | 0.332/0.344| 0.467/0.485|
> |pos: ['good', 'high'], neg: ['poor', 'low'] | 0.539/0.600 | 0.684/0.673 | 0.299/0.324 | 0.565/0.568 | 0.660/0.721 | 0.306/0.330 | 0.414/0.422| **0.495/0.520 (best ensemble)**|
> |pos: ['good', 'fine'], neg: ['poor', 'bad'] | 0.495/0.558 | 0.596/0.581 | 0.264/0.307 | 0.521/0.548 | 0.640/0.691 | 0.270/0.270 | 0.435/0.449| 0.460/0.486|
> |pos: ['good', 'high', 'fine'], neg: ['poor', 'low', 'bad'] | 0.541/0.600 | 0.632/0.617 | 0.286/0.316 | 0.570/0.577 | 0.664/0.719 | 0.301/0.318 | 0.416/0.429| 0.487/0.511|
> |pos: ['good', 'high', 'excellent', 'fine'], neg: ['poor', 'low', 'bad'] | 0.519/0.581 | 0.616/0.601 | 0.274/0.302 | 0.583/0.590 | 0.674/0.723 | 0.316/0.330 | 0.421/0.432| 0.486/0.508|
>
> ##### 5. InternLM-XComposer-VL (improves 0.1%)
>
> | **Model Name**| KoNIQ-10k| SPAQ| LIVE-FB| LIVE-itw| AGIQA-3K| CGIQA-6K| KADID-10K| average|
> | -| -| -| -| -| -| -| -| -|
> |pos: ['good'], neg: ['poor'] | 0.564/0.615 | 0.730/0.750 | 0.360/0.416 | 0.612/0.676 | 0.732/0.775 | 0.243/0.265 | 0.546/0.572| *0.541/0.581 (best single)*|
> |pos: ['fine'], neg: ['bad'] | 0.546/0.597 | 0.720/0.736 | 0.341/0.389 | 0.626/0.671 | 0.681/0.708 | 0.213/0.227 | 0.494/0.479| 0.517/0.544|
> |pos: ['high'], neg: ['low'] | 0.543/0.590 | 0.704/0.720 | 0.331/0.372 | 0.612/0.656 | 0.716/0.755 | 0.223/0.251 | 0.490/0.500| 0.517/0.549|
> |pos: ['good', 'high'], neg: ['poor', 'low'] | 0.564/0.613 | 0.723/0.743 | 0.354/0.405 | 0.621/0.676 | 0.734/0.775 | 0.238/0.264 | 0.522/0.546| 0.537/0.575|
> |pos: ['good', 'fine'], neg: ['poor', 'bad'] | 0.573/0.626 | 0.735/0.755 | 0.366/0.420 | 0.629/0.687 | 0.732/0.771 | 0.236/0.260 | 0.531/0.551| **0.543/0.581 (best ensemble)**|
> |pos: ['good', 'high', 'fine'], neg: ['poor', 'low', 'bad'] | 0.571/0.621 | 0.728/0.748 | 0.360/0.410 | 0.629/0.683 | 0.734/0.773 | 0.236/0.261 | 0.521/0.538| 0.540/0.576|
> |pos: ['good', 'high', 'excellent', 'fine'], neg: ['poor', 'low', 'bad'] | 0.570/0.620 | 0.728/0.747 | 0.356/0.404 | 0.637/0.684 | 0.727/0.763 | 0.237/0.259 | 0.507/0.529| 0.537/0.572|
>
> We would like to again thank the reviewer for providing this solid strategy, which brings **only negligible** additional cost but can improve the accuracy of MLLMs on the image quality assessment (IQA) task. We believe this can help improve our work by including this as additional discussions, and contribute to the IQA community.

---

### Official Review · Reviewer_gz7s · 2023-11-04

**Soundness:** 3 good
**Presentation:** 3 good
**Contribution:** 3 good
**Rating:** 6
**Confidence:** 5

**Summary:**

This work introduces a benchmark for low-level perception and understanding. They introduce three tasks for assessing the MLLM. The main focus is low-level, thus they build an evaluation bench from various aspects (low-level attributes, visual distortions, etc.). The experiments are extensive and comprehensive.

**Strengths:**

1. The motivation is sufficient, and this benchmark is specifically designed for low-level tasks rather than a holistic evaluation of general abilities.
2. The tasks consist of classification and description, as well as probability-based quantitative evaluation, making it well-organized for evaluating the low-level abilities of the current MLLM.
3. The experiments are extensive and comprehensive, which are helpful in diagnosing the strengths and shortcomings of current MLLMs.

**Weaknesses:**

The main concern is about the trustworthiness of the evaluation.

1. As the ChatGPT also suffers from hallucination, how can we ensure the reliability and confidence of the GPT-assisted evaluation?
2. What is the difference between the proposed metric and perplexity (only considering good/bad)? Is the PPL equivalent to this new metric?
3. As current models typically leverage large amounts of data, how can we avoid contamination of the evaluation dataset in the training set?

**Questions:**

As questions in weaknesses

---

> ### Author Response · Authors · 2023-11-11
> **(Thread 1/3) Thank you for your recognition and questions. Here are some clarifications to question 1.**
>
> [Thread 1/3]
>
> ## General Response
>
> We would like to sincerely thank the reviewer for recognition on our motivation, organization, and experimental soundness. We also sincerely appreciate the profound questions as raised by the reviewers. We also agree that improving the **truthworthiness** of the evaluation is also very **important**, and we would like to carefully respond to the questions as follows.
>
> ## Question 1 (Q1)
>
> As the ChatGPT also suffers from hallucination, how can we ensure the reliability and confidence of the GPT-assisted evaluation?
>
> ## Answer to Q1
>
> We are also concerned with the accuracy of GPT-assisted evaluations. In our benchmark, the GPT-assisted evaluation is employed in both the perception (A1) and description (A2) task evaluation.
>
> ### 1.1. Reliability of GPT-assisted Evaluation for the Perception task (A1)
>
> For the perception task, the GPT-assisted evaluation helps determines the correctness of the MLLMs’ responses to the multi-choice questions (MCQ). This is because some MLLMs might not respond as the required format of multiple choice answer. For example, they might respond to the prompt of “How’s the level of blur in the image? Choose between the following answers: A. Some blur, B. Not blurry at all, C. Very blurry.’” with “The image is severely blurry”. The answer is correct (C is the right choice) but not in the required format.  GPT can help us precisely determine the correctness for such responses.
>
> As answering MCQ can only be correct or incorrect without an intermediate state, this judgment is relatively simple and has **fairly objective standards**, that we can check the correctness of GPT evaluation. We notice that a single turn evaluation only results in 93.2% accuracy, which is not reliable enough. To solve this, we propose a **5-round voting protocol**, that lets GPT to judge 5 times and take the majority vote as the evaluation result. It improves the evaluation accuracy to **98.4%**, which can support the general conclusions in the Q-Bench.
>
> We hope this can clarify the concerns of GPT evaluation reliability on the perception task (A1).
>
>
> ### 1.2. Discussion on GPT-assisted Evaluation for the Description task (A2)
>
> For the description task (A2), we have to acknowledge that judging whether a description matches the gold description is a somewhat subjective process. Even when evaluated by humans, the scores rated for the MLLM descriptions are subject to individual differences.
>
> Similar as the previous task, we also employ the **5-round** GPT-assisted evaluation protocol, which average the score of five GPT-rated scores ($s_i, i=[0,1,2,3,4]$), each in one of the integers [0,1,2], into a floating point score ($s$).
>
> $$ s = \frac{\sum_{=0}^{4} s_i}{5} $$
>
> We understand this is an imperfect way to evaluate the low-level descriptions (that may reduce hallucinations but cannot eliminate), yet this could be the most reliable and reproducible way at present. Guided by your valuable feedback, we will continue to explore how to design a more reliable evaluation protocol for the low-level visual description task in our follow-up works.

---

> ### Author Response · Authors · 2023-11-11
> **(Thread 2/3) Thank you for your recognition and questions. Here are some clarifications to question 2.**
>
> [Thread 2/3]
>
> ## Question 2 (Q2)
>
> What is the difference between the proposed metric and perplexity (only considering good/bad)? Is the PPL equivalent to this new metric?
>
> ## Answer to Q2
>
> Thank you for this question regarding the proposed quantitative evaluation (assessment task, A3). In short, the main difference between the proposed metric and *common perplexity (PPL) metric* is that our metric is **specially constrained** to provide **quantitative scores** for images.
>
> ### Main Difference: the Constraints
>
> Considering the **evaluation setting**, we use MLLMs to generate responses starting with "The quality of the image is" to prompt them to answer related to image quality. The goal is to create a straightforward, quantifiable score, and avoid the non-quantifiable free-format sentences (e.g., "The quality of the image is blurry, with the vehicle's details totally invisible.").
>
> To achieve this setting, we pose two constraints to the generation process:
>
> 1. Prediction Limitation: Only one subsequent token is predicted.
> 2. Output Restriction: The output is limited to a pair of opinion-related antonyms (e.g., "good" and "poor"), to avoid distractions from non-quality-indicative tokens like "very", "not", "blurry".
>
> These constraints help us to **limit** an MLLM into a binary classifier, which helps us to obtain a score on visual quality, without requiring to train a new classification/regression head for MLLMs.
>
> ### Equivalence Under the Constraints
>
> Given the constraint, we denote the constrained logits as $\hat l = [l_{good}, l_{poor}]$, and then the constrained probability vector is as $\hat p = [\frac{e^{l_{good}}}{e^{l_{good}}+e^{l_{poor}}},\frac{e^{l_{poor}}}{e^{l_{good}}+e^{l_{poor}}}]$.
> Then, the **proposed metric is to** use $\hat p_0$ directly as the output quality score (as per Eq. 1 in our draft), and the PPL of label 'good' under our constraints is $-\log \hat p_0$. This is the negative logarithmic value of our proposed output quality score.
>
> We opt for the $\hat p_0$ score since it falls within the [0,1] range and is monotonously related to the PPL-based quality score. This makes it more intuitive (*image quality scores are bounded, and better images have higher scores*) and aligned with our evaluation needs.
>
> ### Summary
>
> In conclusion, we leverage the **text generation** ability of MLLMs and use constraints to turn it into a binary classifier, which is specially designed for this task. The scores we obtained can be considered as nearly equivalent to PPL metric **under the constraints**.
>
> We hope the proposed metric, as *a constrained set* perplexity-like metric, can bring the insights of *using LM losses as metrics* into new tasks.

---

> > ### Author Response · Authors · 2023-11-11
> > **A shorter explanation for Question 2.**
> >
> > In shorter words, this proposed metric can be considered nearly as a **close-set** PPL, under a **new LM head** which is only able to predict two words (as a partial copy of the original MLLM head). We do this mainly because MLLMs are not specially tuned for quality assessment, so we need to manually constrain them to avoid unwanted outputs as distractions of our evaluation. We hope this can help clarify the difference between our strategy and common full-dictionary perplexity metrics.

---

> ### Author Response · Authors · 2023-11-11
> **(Thread 3/3) Thank you for your recognition and questions. Here are some clarifications to question 3.**
>
> [Thread 3/3]
>
> ## Question 3 (Q3)
>
> As current models typically leverage large amounts of data, how can we avoid contamination of the evaluation dataset in the training set?
>
> ## Answer to Q3 / Statements on Data Contamination
>
> Same as the reviewer's opinion, we also believe that avoiding data leakage in a benchmark is crucial. Here as follows is a brief statement  on potential data leakage in our benchmark.
>
> *We will add this report in the appendix of the revised paper.*
>
> In short, our benchmark contains three tasks, where the first two tasks (A1: perception and A2: description) are evaluated with **our own datasets** proposed with the paper. For these two tasks, the questions, answers, or low-level descriptions in the two datasets are not seen by any existing MLLMs. We will continue to keep the labels in half of LLVisionQA and full of LLDescribe private, to avoid being added into the training sets of any MLLMs. We hope that this measure will allow the Q-Bench to have long-term significance as an indicator of low-level visual abilities.
>
> For the third task, (A3) assessment, the situation is a bit more complicated. We will discuss open-source models and close-source models separately.
>
> For open-source models as tested, almost all of them have provided their technical reports, and as far as we know, **no** image quality assessment (IQA)  dataset has participated in the **multi-modality training stages** of them. While knowledge about image quality assessment might be injected to them (*e.g. a blurry image is a low quality image*) during their **pure-language training stages**, we think this should not be regarded as data contamination for image quality assessment (IQA), because the images cannot be seen by a language model. Instead, they are important knowledge that helps them to better link particular visual attributes (*blur*) to human opinions (*quality*), which is one of the reasons why we would like to explore MLLMs for these tasks.
>
> For close-source models, e.g. GPT-4V (mainly discussed in response to R3), we are not sure about their data contamination situation on existing datasets (i.e. IQA datasets for task A3 assessment). However, as they are also not included in the respective comparisons in task A3, this will not cause potential data contamination issues in the Q-Bench.
>
> Considering the evidences above, we conducted the results of Q-Bench *at present* are not affected by known contaminations We will continue to pay attention to future data contamination issues and, in the future, differentiate between models on the leaderboard based on the potential risks of these issues.

---

### Author Response · Authors · 2023-11-11
**Thank you for your efforts and valuable suggestions!**

We would like to sincerely express our gratitude to all reviewers for the general recognition on our work, as all reviewers recommend towards acceptance for this work.

More importantly, we also notice that there are several points that reviewers would like us to clarify or revise to further improve our work, and we reply to them in the response boxes below. We have also added all experiments as requested, including a better IQA strategy, and additional results for more open-source MLLMs, and GPT-4V.

Moreover, we also add a special experiment to invite human to do the MCQs in Q-Bench A1 (low-level perception task), as a *oracle*-like competitor for open-source MLLMs and GPT-4V. We hope this can make this benchmark more solid.

At present, we are adding the experiments and clarifications independently in the response boxes, and we will upload an updated version of our draft to consolidate all these revised parts soon (during the rebuttal period). We hope to hear your further questions and suggestions!

Again, thank you all for your efforts, opinions, and valuable suggestions!

Best Regards,
Authors Team of the Q-Bench

---

### Author Response · Authors · 2023-11-20
**Thank you for your efforts! The revised paper is uploaded.**

Dear reviewers:

Thank you for efforts again! Your suggestions are valuable to improving this paper.

In the past week, in addition to replying the official reviews as follows, we have organized all the changes as advised, and responses to the key concerns of reviewers to the paper, into the revised manuscript. Now, we upload the updated version of the paper (PDF file), which integrates all the changes that have been made. We sincerely hope this can be an improved version of this paper that better contributes to this community.

Best Regards,
Authors Team of the Q-bench

---

> ### Comment · Reviewer_rhgp · 2023-11-22
>
> Thank you very much for the extremely thorough responses and the revision of the paper. The quality of the paper has improved considerably, thus, I'd like to raise the rating to Accept. Great job!

---

### Meta-Review · Area_Chair_nUYU · 2023-12-06

**Metareview:**

This paper proposes Q-Bench, a novel benchmark for evaluating the low-level visual perception and understanding capabilities of general-purpose multimodal models.  Three reviewers provided ratings of 8, 6, 8.  All reviewers find the novelty of the benchmark, specifically its focus on evaluating low-level visual understanding capabilities, to be a good contribution.  Other positive points include the motivation, comprehensive experiments, and clear presentation.  Some concerns included lack of more recent baselines including GPT4V, questions surrounding the reliability of using GPT for assessment, and data contamination.  These concerns were adequately addressed during the rebuttal period and revised manuscript.  The paper, rebuttal, discussion, and author messages were carefully discussed among the ACs, and the ACs agree that this is a solid paper worthy of publication.

**Justification For Why Not Higher Score:**

Based on the points made above, this paper is worthy of a spotlight presentation, as it will help further progress in the area of multimodal models.

**Justification For Why Not Lower Score:**

All reviewers and ACs agree that the paper should be accepted.

---

### Decision · Program_Chairs · 2024-01-16

Accept (spotlight)